# Longitudinal multi-omics analysis of host microbiome architecture and immune responses during short-term spaceflight

Maintenance of astronaut health during spaceflight will require monitoring and potentially modulating their microbiomes. However, documenting microbial shifts during spaceflight has been difficult due to mission constraints that lead to limited sampling and profiling. Here we executed a six-month longitudinal study to quantify the high-resolution human microbiome response to three days in orbit for four individuals. Using paired metagenomics and metatranscriptomics alongside single-nuclei immune cell profiling, we characterized time-dependent, multikingdom microbiome changes across 750 samples and 10 body sites before, during and after spaceflight at eight timepoints. We found that most alterations were transient across body sites; for example, viruses increased in skin sites mostly during flight. However, longer-term shifts were observed in the oral microbiome, including increased plaque-associated bacteria (for example, *Fusobacteriota*), which correlated with immune cell gene expression. Further, microbial genes associated with phage activity, toxin–antitoxin systems and stress response were enriched across multiple body sites. In total, this study reveals in-depth characterization of microbiome and immune response shifts experienced by astronauts during short-term spaceflight and the associated changes to the living environment, which can help guide future missions, spacecraft design and space habitat planning.

The sources and health impacts of spaceflight-associated microbiome shifts are an open, yet important area of study. Microbes play manifold roles in human physiology; understanding the complex interplay between the space environment and host-microbiome composition is critical. This is especially true with the recent proliferation of commercial spaceflight missions and increased space tourism: individuals with increasingly diverse medical histories will travel into space and to the Moon (for example, dearMoon)[1], and these crews will also carry a more complex history and range of microbiome states (for example, recent antibiotic usage). In this new age, astronauts can be immunocompromised, cancer survivors, elderly or have other health profiles that put them severe other severe outcomes, especially relative to previous NASA, ESA, JAXA and ROSCOSMOS missions[2].

Microbes are already associated with many spaceflight-specific health indications. In microgravity, many individuals experience gastrointestinal discomfort (that is, constipation), which is heavily linked to gut microbiome composition[3–7]. The skin barrier is disrupted and often inflamed, allowing potential invasion of pathobionts or otherwise inflammatory microorganisms[8–12]. Although the mechanisms are not entirely understood, the immune system experiences suppression during spaceflight, leading to inflammation or a 'reactivation' of latent infections, such as herpes viruses[13–17]. As a result, identifying the sources and impacts of microbiome changes as a function of spaceflight will be essential for the development of microbiome-targeted, spaceflight-relevant diagnostics and therapeutics.

✉e-mail: chm2042@med.cornell.edu

Microbial physiology, genetics and community composition are also dramatically affected by the space environment, probably due to the stressors of microgravity and radiation[18–20]. These changes, taken together, alter the nature of microbial communities and, therefore, their cumulative impact on the host[21]. We recently documented the 'International Space Station (ISS) effect', in which organisms on the ISS exhibit increasing resistance to antibiotics over time, despite not having been exposed to them in the first place[22]. Many Biosafety Level 2 organisms, including *Haemophilus influenzae, Klebsiella pneumonia, Salmonella enterica, Shigella sonnei* and *Staphylococcus aureus*, have been observed to exhibit ecological succession in the environment of the ISS, demonstrating the propensity of the space environment to select for specific community compositions and gene content[23,24].

Early studies in aerospace medicine have indicated that the microbiomes of humans and the built environment shift as a function of spaceflight[25]; however, there are many open questions regarding spaceflight's microbiome architecture: the totality of detectable flight-associated compositional and expression shifts in the set of all bacteria, viruses and microbial genes in the host and their surrounding environment (Glossary/Supplementary Table 1). Also, the proportion of organisms acquired from other crew members, versus from the environment, remains unclear, and the transience of microbiome changes post-flight remains opaque. Notably, the metatranscriptomic activity of human-associated microbes in response to flight is completely absent. These questions predominantly remain because previous studies have been hampered by (1) limited sample sizes, (2) a lack of longitudinal data and (3) a focus on single sequencing modalities (that is, amplicon sequencing or only DNA profiles).

To interrogate microbiome community activity in spaceflight, we recently executed a longitudinal, multi-omic (metagenomics, metatranscriptomics, single-cell immunome) sampling study of the SpaceX Inspiration4 mission (i4)—the first all-civilian commercial spaceflight. Over a 6-month window, the crew collected environmental (that is, from the Dragon capsule), skin (n = 8 sites), nasal and oral swabs at eight timepoints before, during and after a 3-day, high-elevation (590 km) mission in orbit, as well as peripheral blood mononuclear cells (PBMCs) before and after spaceflight (n = 3 per flight window). We focused on expression and abundance shifts and their relationship to host immune status as a function of spaceflight. Our results yield a standardized approach for temporally monitoring microbial exposomic changes as a function of spaceflight and, in total, characterize the microbiome architecture[26] of biomedically relevant taxa that are activated or repressed during short-term spaceflight.

## Results

### The human microbiome is altered in short-term spaceflight

The i4 crew collected a microbiome dataset spanning eight timepoints: three before flight, three after flight and two during flight. In total, we sequenced 385 metagenomic and 365 metatranscriptomic swabs comprising ten body sites representing the oral, nasal and skin microbiomes (n = 750 samples, Fig. 1a), plus eight stool samples (from two participants before and after flight). Locations inside the Dragon capsule were swabbed twice in flight and also before spaceflight (a separate capsule was utilized for crew training).

We used a diverse set of short-read alignment and de novo assembly approaches to estimate the microbial community taxonomic and functional composition of our dataset (Extended Data Fig. 1, Methods and Supplementary Figs. 1–7). We queried whether the conditions of short-term spaceflight (potentially including, for example, microgravity, radiation or altered dietary and cleaning habits) altered overall bacterial and viral community composition and expression consistently across the crews. Via a linear mixed effect (LME) modelling approach, we executed a microbiome-association study (MAS), computing associations for each taxonomic rank and

classifier between flight and the abundance of (1) bacterial species, (2) viral genera and non-redundant proteins. We grouped false discovery rate (FDR) significant (q-value < 0.05) features into four categories: transiently increased in-flight, transiently decreased in-flight, persistently increased in/after flight and persistently decreased in/after flight. We additionally fit generalized linear models (GLMs) alongside LMEs and identified the two approaches to be generally concordant (Extended Data Fig. 2).

In total, we observed a predominantly transient restructuring of the oral, nasal and skin microbiomes as a function of flight (Fig. 1b–c and Extended Data Fig. 3). Across all ten sites swabbed and regressed, over 821,337 associations were statistically significant (adjusted P < 0.05) and grouped into one of the four categories of interest. These comprised 314,701 distinct microbial features: 792 were viral, 767 were bacterial and the remaining were microbial genes. The majority (73.5%) of significant and categorized features were transiently increased in abundance, yet 24.6% were transiently depleted during flight, and 0.6% and 1.1% of features appeared to continually increase or decrease (respectively) following the crew's return to Earth across multiple timepoints. Transient shifts were more dramatic than persistent ones. The limited persistence of changes indicates that, while microbial communities may restructure in space, the relative abundance of altered organisms, as well as their gene expression, generally reset upon returning to Earth. Despite these changes, we note that this analysis alone does not indicate the degree to which spaceflight itself, versus other confounding factors such as altered diet, affects the host microbiota.

Different body sites displayed distinct time trends that varied depending on molecular type (gene expression vs relative abundance) and domain of life. Time-dependent shifts were apparent in all body sites. The oral microbiome displayed a restructuring in both relative abundance and bacterial gene expression; 161 bacterial and viral taxonomies were transiently increased, 173 were transiently decreased, 62 were persistently increased and 12 were persistently decreased (Fig. 1c). Alternatively, the skin microbiome demonstrated fewer persistent changes, with 933 transiently increased (metagenomic) taxa across all eight skin sites. The number and direction of altered microbiome features were generally consistent across classification methods (Extended Data Fig. 4), and most taxonomic associations were unique to individual body sites (Extended Data Fig. 5).

### Bacterial and viral shifts during and after spaceflight

We next interrogated the taxonomies of bacterial shifts during spaceflight. The organisms with the strongest effects were distinct across biological modalities; in other words, an increase in gene expression did not necessarily imply the existence of a similar increase in the abundance of DNA ascribed to a given species. This discordance was apparent in the oral microbiome (Fig. 2), for example, where there was almost no overlap between the organisms that altered in terms of relative abundance and those that altered in terms of gene expression.

Specifically, the oral microbiome demonstrated flight-dependent variation in the metatranscriptomic expression of bacteria associated with dental decay and biofilm formation (Fig. 2). Various members of *Fusobacteriota*, a progenitor to gum and tooth disease previously reported as spaceflight-associated, demonstrated an increase either in or after spaceflight[27]. These included *Fusobacterium hwasookii, Fusobacterium nucleatum* and *Leptotrichia hofstadii*. Other oral biofilm species known to aggregate synergistically with *Fusobacterium* species in the mouth were also enriched in and after flight; these included *Streptococcus gordonii A*, multiple *Campylobacter* species and *Actinomyces oris* species[28]. Also, there was a persistent loss in the expression of *Streptococcus oralis* spp. and *Lachnoanaerobaculum gingivalis*, and a transient decrease in *Veillonella* spp. *Alloscardovia omnicolens* was the only organism with a strong, persistent increase in metagenomic levels. We compared the MetaPhlAn4 associations to those identified

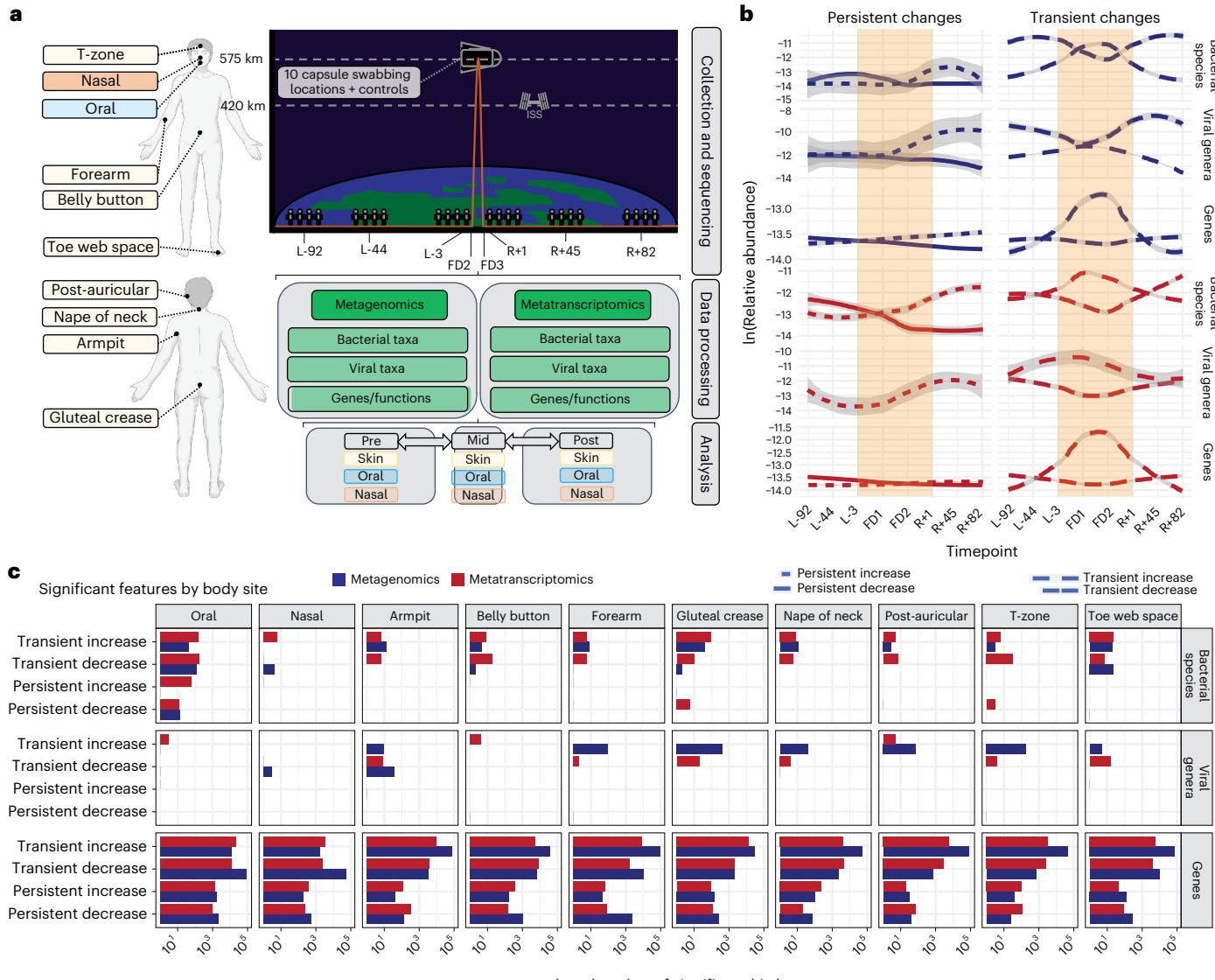

**Fig. 1 | Overview of dataset and summary of changes. a**, Collection and analytic approach. Body swabs were collected from ten different sites, comprising three microbial ecosystems (oral, nasal, skin) around the body at eight different timepoints surrounding launch. These are referred to as L-92, L-44, L-3, FD2, FD3, R+1, R+45, R+82, where 'L' refers to pre-launch, 'FD' corresponds to flight day (that is, in-flight) and 'R' refers to return (that is, post-flight). Following collection and paired metagenomic/metatranscriptomic sequencing, samples were processed with a MAS to extract taxonomic (bacterial or viral) and functional features to determine their changes relative to flight. **b**, The time trajectories of persistently/transiently increased/decreased significant findings, filtering for strong associations (see Methods). Plots with one line had either no significant findings or none that met the filtering criteria. Grey shaded area indicates 95% confidence intervals. The orange shaded area refers to the samples collected while subjects were in orbit. **c**, Significant features by specific swabbing sites.

in the GTDB database and found similar results, especially regarding the overall enrichment of *Fusobacterium* sp. in flight.

Many of the strongest bacterial skin microbiome alterations (Fig. 3) were predominantly metagenomic, as opposed to metatranscriptomic. We hypothesized that this may indicate the acquisition of new but non-transcriptionally active species from the surrounding environment and crew. For example, persistent increases were mostly in the metagenomic content of various gut microbes (for example, *Bacteroides*, *Parabacteroides*, *Blautia*, *Enterocloster*); this may result from altered hygiene habits during spaceflight.

As with the oral microbiome, there was little concordance between metagenomic and metatranscriptomic changes. On the other hand, *Corynebacterium* species (common skin commensals) experienced metatranscriptomic, temporary depletion in-flight, and *Acinetobacter* spp. demonstrated a persistent depletion. These 'typical' skin

microbes (for example, *Corynebacterium*, *Staphylococcus*, *Variovorax*, *Acinetobacter*) underwent changes in metatranscriptomic activity, whereas organisms not universally found on the human skin (for example, *Mesorhizobium* spp., *Prevotella* spp.) tended to experience metagenomic shifts, again indicating the potential acquisition of non-transcriptionally active organisms from different niches.

However, the landscape of viral activity and depletion covered region-specific, prokaryotic- and eukaryotic-targeting viral genera (Fig. 3b). The majority of detectable viral activity comprised phages in the skin microbiome (that is, DNA viruses targeting prokaryotic hosts), and it was concentrated in the gluteal crease. Most viral activity was transiently increased in-flight across diverse lineages. For example, *Uroviricota*, *Cressdnaviricota* and *Phixviricota* shifted across the oral, skin and nasal microbiomes. However, phyla containing biomedically relevant, potential human pathogens also increased,

Oral microbiome shifts

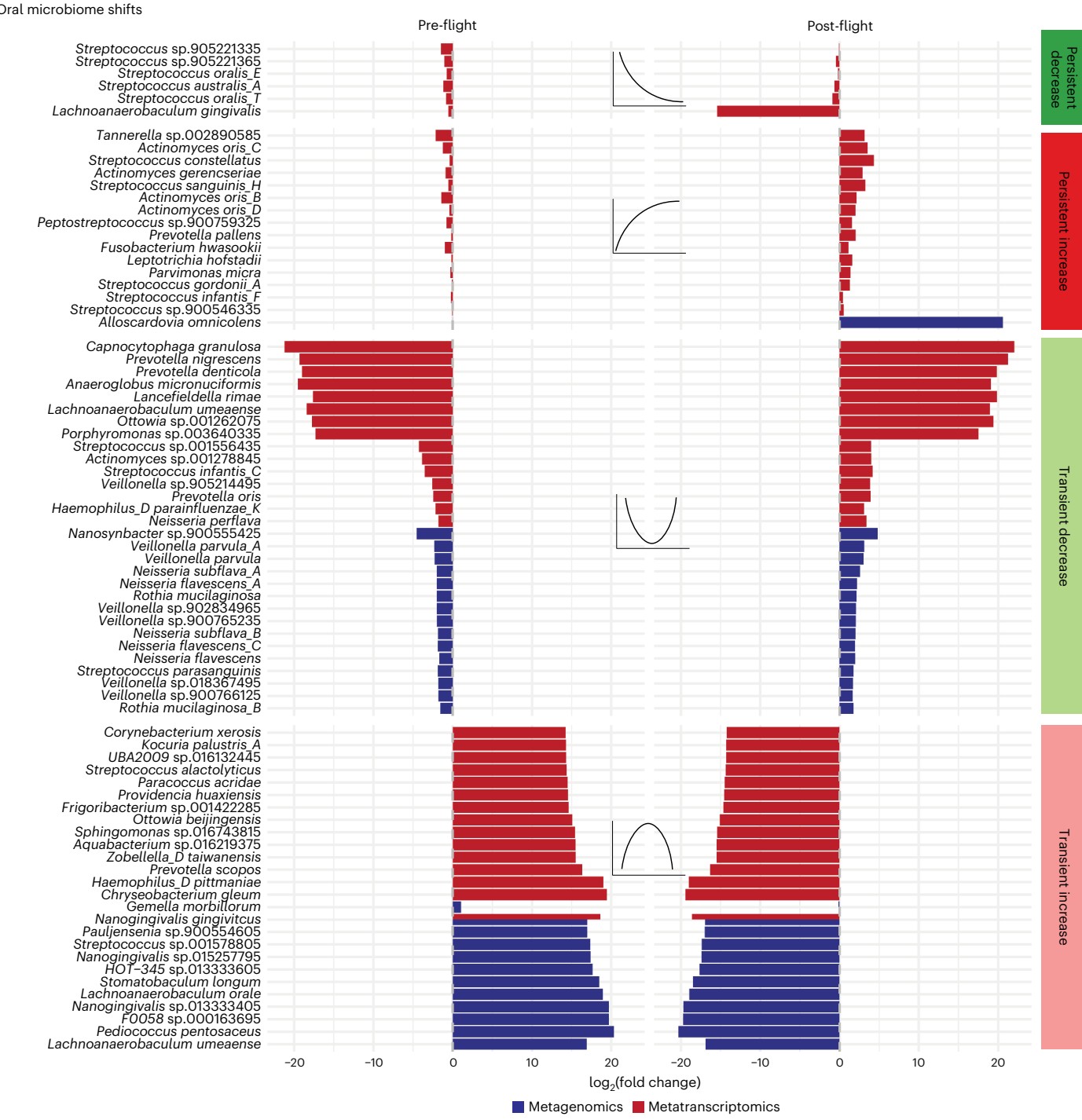

**Fig. 2 | The oral microbiome architecture of spaceflight.** The strongest associations between bacteria and flight for the oral microbiome. $X$ axes are average $\log_2$ fold change ($L_2FC$) of all pre-flight or post-flight timepoints compared to the average mid-flight abundances for a given taxon. Columns correspond to different association categories that are described visually by the example line plots on top of each one. Red bars refer to associations in metatranscriptomic data. Blue bars refer to associations in metagenomic data. Dashed grey vertical lines demarcate an $L_2FC$ of zero. Plotted taxa were selected by ranking significant features in each category and sequencing type (RNA/DNA) by $L_2FC$ and showing up to 15 at once.

including *Kitrinoviricota*, *Artverviricota*, *Nucleocytoviricota* and *Duplornaviricota*.

## A core functional microbial landscape of spaceflight

We next aimed to characterize the consistency with which microbial gene abundances changed across time and body site across 3.6 million non-redundant genes. First, we explored the broad functions of the genes that fell into either the transiently increased or transiently

decreased categories. The increases in DNA content on the skin, as well as decreases in nasal microbiome content, were immediately apparent (Extended Data Fig. 6a, third and first columns, respectively), and the oral microbiome and gluteal crease also underwent large metatranscriptomic increases. Of note, the category that exhibited the greatest fluctuation in genes was 'amino acid transport and metabolism'. In the exposed areas of the skin, such as the forearm, the genes that were changed in this category mostly came from metagenomic data.

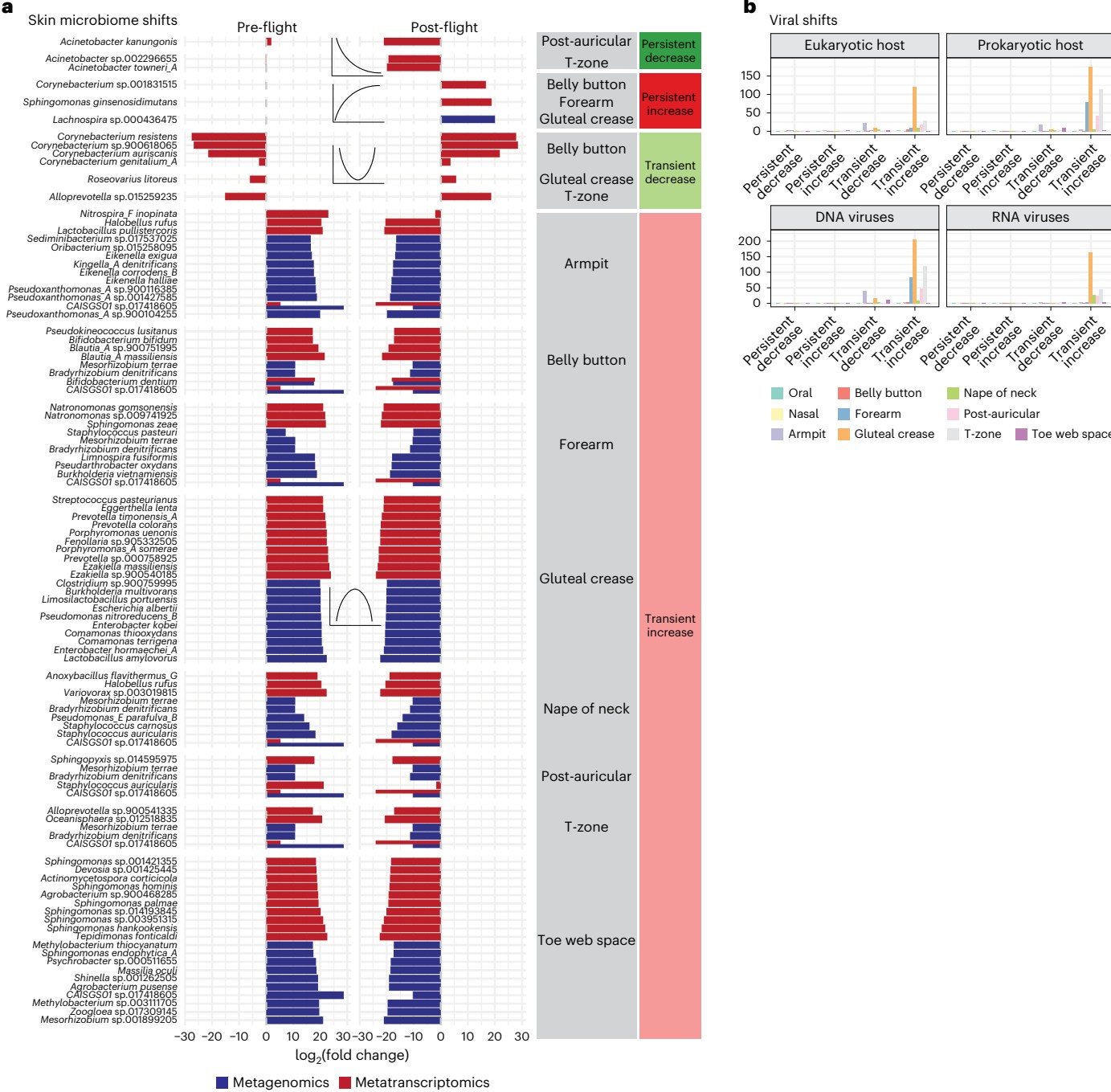

**Fig. 3 | The skin microbiome and viral architecture of spaceflight. a**, The strongest associations between bacteria and flight for the skin microbiome. *X* axes are the average $L_2FC$ of all pre- or post-flight timepoints compared to the average mid-flight abundances for a given taxon. Columns correspond to different association categories that are described visually by the example line plots on top of each one. Dashed grey vertical lines demarcate an $L_2FC$ of zero. Plotted taxa were selected by ranking significant features in each category and sequencing type (RNA/DNA) by $L_2FC$ and showing up to 10 at once. **b**, Host and molecular type of viral genera associated with flight.

In less exposed body sites (that is, oral, gluteal crease), the activity in this category was primarily metatranscriptomic. This may indicate the dramatic degree to which microbial nutrient needs change in-flight, probably from a combination of features ranging from environmental strain transfer, competition and host dietary changes.

The oral, nasal and skin microbiomes demonstrated consistency in the functions that were altered during spaceflight, especially in the metagenomic data. We observed five different categories of proteins of interest enriched among increased features: antibiotic and heavy metal resistance, haem binding/export, lantibiotic-associated genes, phage-associated genes and toxin–antitoxin systems (Extended Data Figs. 6b and 7–9). Lantibiotic biosynthesis again displayed a discordant response between sequencing types; it was decreased in the metagenomic data but increased in metatranscriptomics. Phage proteins, toxin–antitoxin systems, and antibiotic-related/heavy metal pathways increased noticeably across all host niches. As in other spaceflight studies[22,29], we specifically observed an increase in the RelB toxin–antitoxin genes, most notably through metatranscriptomics.

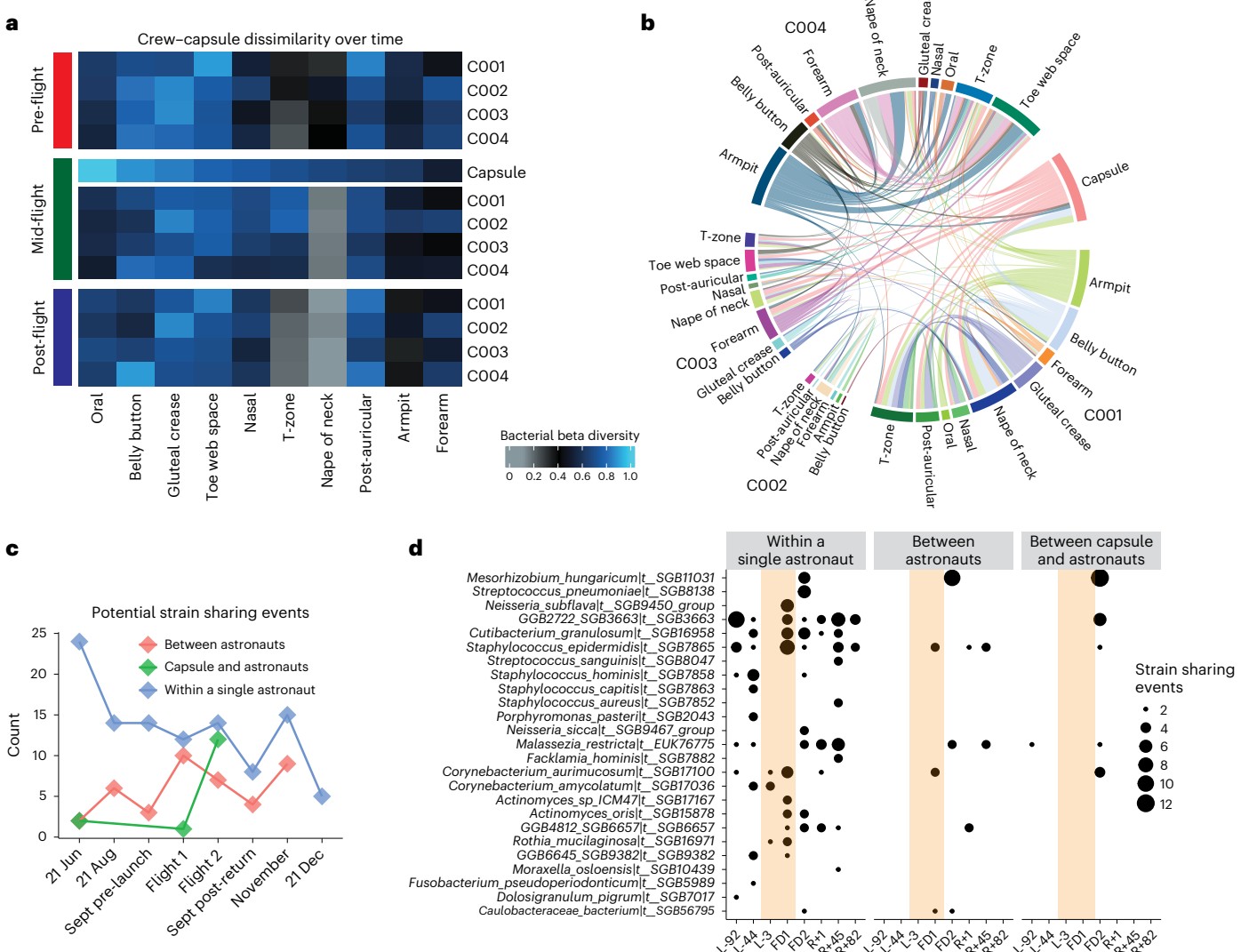

**Fig. 4 | Microbial propagation through the Dragon capsule and the crew.**
**a**, Beta diversities for bacterial metagenomics. Heat map colour corresponds to average beta diversity, with black being the midpoint (0.5), blue being totally dissimilar (1.0) and grey being highly similar (0.0). Columns are hierarchically clustered. The interpretation for a single cell is for the crew member annotated on the right-hand side: the value encoded refers to the dissimilarity of that individual's body site (as indicated on the column) to all other cells in that column (so the capsule and all other crew samples from the same site). For example, the bottom right cell indicates C004's average forearm dissimilarity to all other crew member's forearm swabs. **b**, Strain-sharing events between the crew and the capsule during the mid-flight timepoints. **c**, The number of strain-sharing events across time, where an event is defined as the detection of the same strain between two different swabbing locations. **d**, Organisms with at least two strain-sharing events detected within a given timepoint.

## Microbial similarity between the capsule and crew members

We observed that, on average, bacterial beta diversity appeared to decrease after flight (Fig. 4a), indicating a broad convergence of the crew microbiome. When ranking sites by similarity to the capsule mid-flight (Fig. 4a, from left to right), the beta diversity correlated with the degree of environmental exposure for a given sampling site. For example, the oral microbiome remained highly dissimilar from the microbiome of the capsule and other sites, whereas the forearm microbiome became much more similar to that of the walls of the Dragon capsule and other crew members, which matches the degree of exposure of that body site.

Further, our MAS also indicated that during spaceflight, the composition of the crews' microbiota changed, most notably in the skin niche, although the sources of these alterations were unclear. We hypothesized that these shifts in community composition and the overall increase in microbiome similarity could simply be a result of individuals cohabitating in a tight space; however, a change in gene expression in the oral microbiome, where microbial exchange is probably less likely, could derive from other ecological or other exposure changes such as diet or immune alterations.

Therefore, we next identified shared microbial signatures between individuals and the environment. Specifically, we queried whether host microbiomes converged during and after flight and whether putative microbial exchange occurred within individuals, between individuals, or both within individuals and the capsule, utilizing recently published methods[30] to determine whether strain-level markers could discern the directionality of microbial exchange across environments.

Overall (Fig. 4b), we found that while in flight, marker genes for individual microbial strains were mostly shared between skin microbiome samples from the same individual (and not across individuals). The capsule had variation in overlap with individuals, with most occurring in exposed skin. Moreover, there were more potential shared microbial strains between the capsule and different individuals by the second sampling in-flight (Fig. 4c), indicating an influence of cohabitation

time on migration. Strains identified by StrainPhlAn, such as *Mesorhizobium_hungaricum|t_SGB11031*, identified as present in multiple locations mid-flight (Fig. 4d), were similar in part to those GTDB species identified as increased metagenomically (but not transcriptionally) across exposed skin sites (Fig. 3). Notably, most of these putative sharing between individuals were present after flight, as opposed to before the mission, indicating in-flight transfer. Finally, we note that in this dataset, which contains numerous low-biomass samples, any indications of strain-sharing are difficult to validate (for example, contamination could potentially drive some of the findings), so the following results should be validated in future skin microbiome and aerospace studies.

### Spaceflight microbiome shifts associate with host gene expression

Having mapped the architecture of microbiome changes surrounding spaceflight and identified the source of some of those changes, we next searched for indications of a link between microbiome ecology and the host immune system. To do so, we integrated the observations from our MAS with host immune single-nuclei transcriptome data from PBMCs. Via averaging across single-nuclei sequencing information, we estimated the gene expression of nine host immune cell subpopulations and we computed differentially expressed genes within cell types post-flight using lasso regression to identify candidate relationships between flight-associated, increased microbial features and immune cell subpopulation gene expression. Specifically, we aimed to identify whether metatranscriptomic or metagenomic (that is, cohabitation-derived) changes were more likely to be correlated with immune transcriptome changes.

We observed many putative relationships between host immune cell expression, body site and microbial features (Fig. 5a). Bacterial species, specifically in the oral microbiome, had many metatranscriptomic associations across all cell types. In terms of relative abundance (that is, metagenomics), oral microbes were associated with CD4 T cells, CD8 T cells and CD16 monocytes, which are known for innate immune response against pathogens[31,32]. Skin bacteria had very few associations with immune cells (compared to oral) in both metagenomics and metatranscriptomics. The overall lack of associations between skin bacteria and immune response was interesting, as it indicated that while microbes are potentially acquired during flight (as observed in Fig. 4), these acquired microbes may have limited immediate impact on the host. In other words, there was limited evidence that strain-sharing due to cohabitation drove a strong altered immune state in humans. Conversely, we did observe a limited link in our data between viruses and immune cell expression, with natural killer (NK) cells, CD14 monocytes, dendritic cells and CD16 monocytes showing the most viral associations; these associations were predominantly in the skin microbiome, which may relate to previous observations of increase viral shedding in astronauts.

Next, we examined a subset of microorganisms with expression and abundance changes that correlated with host genes across multiple immune cell types (Fig. 5b). A small group of metagenomically detected viruses were associated with many different immune genes; one genus (Family *Genomoviridae*) targets fungi and was correlated with 13 genes in natural killer cells. The presence of this virus on the skin makes additional sense given that fungi are known skin symbionts. The other associated viruses had unclassified hosts or targeted bacteria.

In the oral microbiome, pathobiont gene expression was associated with immune cell gene expression. *Streptococcus pneomoniae A* had the largest number of genes associated with it; 30/32 genes were found in natural killer cells. *Streptoccocus gordonii A*, which was persistently increased after flight was associated with many different immune cell subtypes (*N* = 32 genes), including CD4 T cells, CD14 monocytes, CD16 monocytes and dendritic cells. The only oral bacterial relative abundance increase during or after flight that was associated with

many immune cell subtypes was in *Gemella morbillorum*. The other oral microbes with the strongest oral associations included other medically relevant organisms, as well as some typical commensals: *Pauljensenia hongkongensis*, *Campylobacter_A concisus_R*, *Actinomyces massiliensis*, *Haemophilus_A parahaemolyticus*, *Leptotrichia_A sp905371725*, *Porphyromonas catoniae* and many *Streptococcus* spp.

The microbial genes (Fig. 5c) associated with the most human genes were detected via both shifts in relative abundance (DNA) as well as expression (RNA). They spanned many different protein annotations, yet there were some commonalities among those that were correlated with many immune cell subpopulations. Most notably, these annotations, across both metagenomics and metatranscriptomics, included transcription factors, cell surface proteins and transporters. Pertinent to our previous results (Extended Data Fig. 6b), the top microbial gene (for example, hemX) in the nasal microbiome was the haem uptake protein IsdC.

## Discussion

In this study, which comprises the largest dataset of spaceflight-associated microbiome data so far, we systematically queried the microbiome architecture of short-term spaceflight. Previous efforts, such as the NASA twins study, have had difficulty mapping microbiome shifts due to small sample sizes, restricted body site sampling (*n* = 3) and limited sequencing modalities[25]. Here we show significant bacterial, viral and gene-level microbiome shifts and their potential relationship to host immune response, which can help inform sampling and monitoring for future missions.

Chief among our findings was that native microbiome shifts were highly correlated with host immune changes. Naturally, a microbial shift can affect the host immune system, or vice versa, without the initial cause being 'space-specific' (that is, due to microgravity or radiation). Dietary factors or other confounders could drive a portion of these effects. Putative convergence in microbiome signatures (Fig. 4a), for example, could be (and probably is) a function of humans residing in close quarters. Moreover, crews have been documented as experiencing immune and viral reactivation[15]; typically, this effect is not attributed solely to cohabitation, and we showed here that species potentially acquired from the environment in flight were not associated with immune cell changes—a topic that has been debated in previous studies[25]. We claim, therefore, that it is unlikely that strain sharing due to close quarters, or even variable sanitation in-flight, could explain the entirety of the link between host immune response and the microbiome. Future manuscripts, of course, could leverage this dataset as well as data from analogue astronaut studies on Earth to test this hypothesis more rigorously.

An additional paper focuses more on the host-side of immune activation[33], reporting specific human genes that seem to be associated with microbial features and integrating additional datasets. However, for completeness, we briefly document here some human genes of interest that were microbiome-associated. By cell type, we documented the most strongly associated genes with microbial features. For bacteria, gene functions were annotated with, for example, long non-coding RNAs (across all cell types), immunoglobulin genes (CD14 monocytes) and interferon regulatory factors. We additionally uncovered associations with specific immune modulatory genes such as CXCL10, XCL1, CXCL8 (immune cell migration), NLRC5, HLA genes, CD1C (antigen presentation/co-stimulation), SLC2A9 (immune cell metabolism), IRF1, NR4A3 and STAT1 (transcription factors that specify immune cell states) that increased across multiple immune cell types (B cells, CD4 T cells, CD8 T cells, CD14 monocytes, DCs, NK cells).

A limitation of our work is its observational nature, which arises from the overall study design and an opportunistic mission. Despite having more samples than all other astronaut microbiome studies combined, this effort still hosts a relatively small crew size (*n* = 4), and we cannot determine from these data alone if an outside effect on the

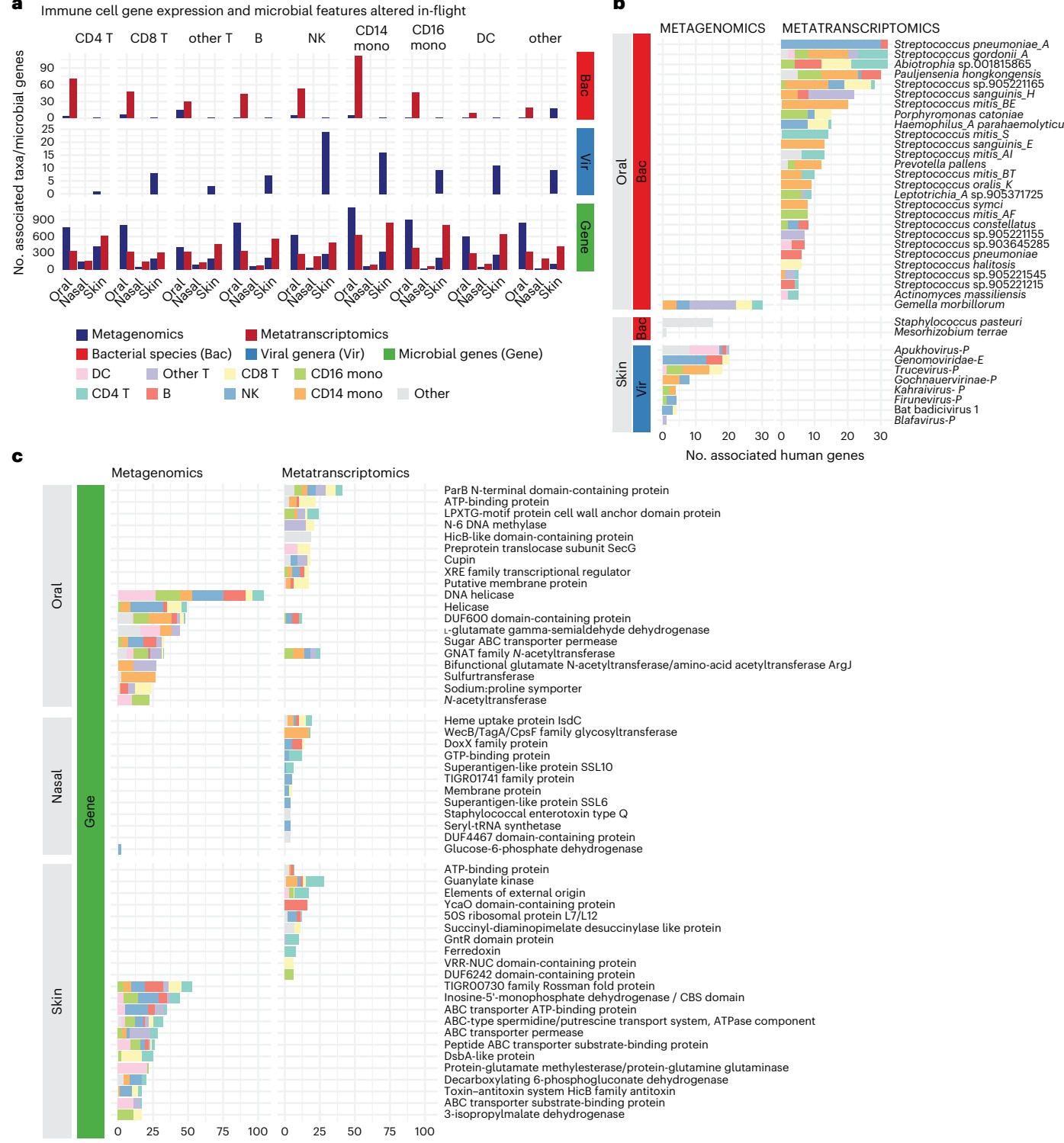

**Fig. 5 | The landscape of potential immune–microbiome associations related to spaceflight. a**, The total number of microbial features, by type, associated with different immune cell subtypes gene expression for those that were long-term increased (left) or decreased (right) after flight. **b**, The spaceflight-associated (increased in abundance or expression) bacteria and viruses that were associated with the greatest number of host genes. Viral genera are labelled 'E' for targeting a eukaryotic host and 'P' for targeting a prokaryote. If no definite host is known, no label was assigned. **c**, The spaceflight-associated microbial genes that were associated with the greatest number of host genes. We sorted for genes within each body site and selected the top 15 with the greatest number of human gene associations. The legend at the bottom of **a** is relevant for all figures where those colours appear.

immune system is altering their abundance or expression or if microbiome ecology may be driving these and similar changes. Given the nascence of the multi-omic space biomedicine (and the difficulty of sample collection), we were limited in this study to simply observing shifts in microbes and, from multi-omic data integration, inferring hypotheses regarding the overall nature of the mid-flight microbe–immune axis.

As a result, there are several opportunities to expand on this work in future studies and missions. Analytically, our lasso-based approach for immune–microbe interaction modelling of immune cell gene expression changes does not inherently allow for statistical inference or account for inter-individual variation. Further, some of our samples had very low biomass, requiring PCR-amplification (18 cycles) for RNA-sequencing data, which can increase duplicate rates of sequences. For this reason, we attempted to take a conservative and systematic modelling approach to our effort. Specifically, (1) we implemented multiple algorithms and compared their concordance, (2) set coverage thresholds for bacterial and viral taxa to filter probable false positives, (3) used multiple, state-of-the-art taxonomic classifiers and compared our findings among all of them, (4) mapped reads across five databases and (5) implemented and compared both generalized linear models and mixed effect models, bearing in mind that the latter can face interpretability challenges with smaller sample sizes. Additional modelling strategies, including network analyses[34], could be implemented in addition to those we have tested here. For example, recently developed network models or methods for controlling false positive rates in compositional data could be potentially useful[34,35].

We additionally used 76 negative controls to attempt to avert false positive signals, which can stem from contamination and the kitome. Depending on their aim, future studies should alter collection methods to increase the amount of biomass collected during sampling (for example, using one swab for multiple skin sites) or examine relatively unbiased methods of amplification[36]. In addition, they should encourage more detailed reporting on diet and cleaning methods (for example, wet wipes) to adjust for potential confounders introducing foreign microbial DNA into the host.

Further, in this study, we attempted to measure viral shifts as a function of flight. Measuring viral abundances in metagenomic and (particularly) metatranscriptomic data is extremely challenging. First, the decontamination process we used to remove environmental contaminants was not designed for organisms as ubiquitous and with such short, diverse genomes as viruses. Second, high-resolution taxonomic classification of viruses is non-trivial. We observed, for example, many spaceflight-associated viruses with unusual names, often mapping to viral species not typically found in or on humans. Simply put, the viral universe is so vast that these alignments may represent both read alignments and as well as spurious read mapping (and databases lacking the strains that are truly present). While our benchmarking efforts (Extended Data Fig. 10) increased our confidence in our results, the viral taxonomic and functional mapping field is an area that will benefit the most from improved methods in the future.

Additional experiments and missions can further test a microbiome-derived theory of spaceflight-associated immune changes. In addition to stress-testing our findings and increasing sample sizes, future spaceflight studies should consider several enhancements. For instance, they should compare sequestered ground controls to discern differences between space-driven and proximity-driven immune shifts. In addition, future efforts should design experiments that enable a deeper view into the causality of microbe–immune associations. Exploring some of these hypotheses through animal or organoid models could be valuable, as well as comparison to large control cohorts.

In total, spaceflight microbiome studies are hyperbolic extensions of human exposome research. They capture a group of effectively immunocompromised individuals who share a self-contained environment that does not undergo microbial exchange with the outside world. Since these studies are rare, the range of immune system dynamics is just beginning to be explored. Overall, we describe here data and methods to map the axes of host–microbe–environment interaction, such that these observations and hypotheses can be tested and even modulated in future studies. Indeed, the increased access to space guarantees more opportunities to study astronauts, their microbiomes and their spacecraft while also motivating a strong health and medical impetus to plan for future missions.

## Methods

### Informed consent and ethics approval
This study was completely in accordance with appropriate ethics guidelines. All participants consented at an informed consent briefing at SpaceX (Hawthorne, California), and samples were collected and processed under the approval of the institutional review board at Weill Cornell Medicine, under Protocol 21-05023569. All crew members provided written informed consent for data and sample sharing.

### Sample collection, extraction and sequencing
We sequenced analysed samples from human skin, oral and nasal environmental swabs before, during and after a 3-day mission to space. This dataset comprised paired metagenomic and metatranscriptomic sequencing for each swab. A total of 750 samples were collected in this study by the four crew members of the SpaceX Inspiration4 mission. The samples were taken from 10 body sites (Fig. 1a) across 8 collection points (3 pre-launch, 2 mid-flight and 3 post-flight) between June 2021 and December 2021. The crew additionally collected 20 samples from multiple Dragon capsules from 10 different locations. We note that some crew members (two adult male, two adult female) were using wet wipes (UPC, 036000317985) to bathe themselves in-flight in between swabbing; however, not every crew member did so, and SpaceX did not require this to be a consistent protocol among the crew. Wet wipes used by the crew were neither reused nor shared, which should limit any influence of this confounding variable. No statistical methods were used to predetermine sample sizes but our sample sizes are greater than any previous publication in this field.

The crew were each provided sterile Isohelix Buccal Mini Swabs (Isohelix, MS-03) and 1.0 ml dual-barcoded screw-top tubes (Thermo Scientific, 3741-WP1D-BR/1.0 ml) prefilled with 400 μl of DNA/RNA Shield storage preservative (Zymo Research, R1100). Following sample collection, swabs were immediately transferred to the barcoded screw-top tubes and kept at room temperature for less than 4 days before being stored at 4 °C until processing. Additional descriptions of the sample collection and sequencing methods are available in companion publications[37]

DNA, RNA and proteins were isolated from each sample using the QIAGEN AllPrep DNA/RNA/Protein kit (QIAGEN, 47054) according to manufacturer protocol, yet omitting steps one and two. To lyse biological material from each sample, 350 μl of each sample was transferred to a QIAGEN PowerBead tube with 0.1 mm glass beads and secured to a Vortex-Genie 2 using an adapter (1300-V1-24) before being homogenized for 10 min. Of the subsequent lysate, 350 μl was transferred to a spin-column before proceeding with the protocol. Concentrations of the isolated DNA, RNA and protein for each sample were measured by fluorometric quantitation using the Qubit 4 fluorometer (Thermo Fisher, Q33238) and a corresponding assay kit. The Qubit 1Xds DNA HS Assay kit was used for DNA concentration (Q33231) and the RNA HS Assay kit (Q32855) was used for RNA concentration.

For shotgun metagenomic sequencing, library preparation for Illumina NGS platforms was performed using the Illumina DNA FLEX Library Prep kit (20018705) with IDT for Illumina DNA/RNA US indexes (20060059). Following library preparation, quality control was assessed using a BioAnalyzer 2100 (Agilent, G2939BA) and the High Sensitivity DNA assay. All libraries were pooled and sequenced on an S4 flow cell of the Illumina NovaSeq 6000 Sequencing System with 2 × 150-bp paired-end reads.

For metatranscriptomic sequencing, library preparation and sequencing were performed at Discovery Life Sciences (Huntsville, Alabama). The extracted RNA went through an initial purification and cleanup with DNase digestion using the Zymo Research RNA Clean & Concentrator Magbead kit (R1082) following the

manufacturer-recommended protocol on the Beckman Coulter Biomek i5 liquid handler (B87583). Following cleanup, ribosomal RNA reduction for RNA-seq library reactions was performed using the New England Bioscience NEBnext rRNA Depletion kit (Human/Mouse/Rat) (E6310X), and libraries were prepared using the NEBnext Ultra II Directional RNA Library Prep kit (E7760X) with GSL 8.8 IDT Plate Set B indexes. Following library preparation, quality control was assessed using the Roche KAPA Library Quantification kit (KK4824). All libraries were pooled and sequenced on an S4 flow cell of the Illumina NovaSeq 6000 Sequencing System with 2 × 150-bp paired-end reads.

For faecal collection, all participants were provided with DNA Genotek OMNIgene-GUT (OM-200) kits for gut microbiome DNA collection. Each participant was instructed to empty their bladder and collect a faecal sample free of urine and toilet water. From the faecal specimen, each participant used a sterile single-use spatula, provided by the OMNIgene-GUT kit, to collect the faeces and deposit it into the OMIgene-GUT tube. Once deposited and sealed, the user was instructed to shake the sealed tube for 30 s to homogenize the sample and release the storage buffer. All samples from each timepoint were stored at room temperature for less than 3 days before storing at −80 °C long term. Faecal samples collected using the OMNIgene-GUT kit are stable at room temperature (15–25 °C) for up to 60 days.

DNA was isolated from each sample using the QIAGEN PowerFecal Pro DNA kit (51804). OMNIgene-GUT tubes were thawed on ice (4 °C) and vortexed for 10 s. Then, 400 µl of homogenized faeces was transferred into the QIAGEN PowerBead Pro tube with 0.1 mm glass beads and secured to a Vortex-Genie 2 using an adapter (1300-V1-24) before being homogenized at maximum speed for 10 min. The remainder of the protocol was completed as instructed by the manufacturer. The concentration of the isolated DNA was measured by fluorometric quantitation using the Qubit 4 fluorometer (Thermo Fisher, Q33238), and the Qubit 1Xds DNA Broad Range Assay kit was used for DNA concentration (Q33265).

For shotgun metagenomic sequencing, library preparation for Illumina NGS platforms was performed using the Illumina DNA FLEX Library Prep kit (20018705) with IDT for Illumina DNA/RNA US indexes (20060059). Following library preparation, quality control was assessed using a BioAnalyzer 2100 (Agilent, G2939BA) and the High Sensitivity DNA assay. All libraries were pooled and sequenced on the Illumina NextSeq 2000 Sequencing System with 2 × 150-bp paired-end reads.

### Sample quality control

All metagenomic and metatranscriptomic samples underwent the same quality control pipeline before downstream analysis. Software used was run with the default settings unless otherwise specified. The majority of our quality control pipeline makes use of bbtools (v.38.92), starting with clumpify (parameters: optical = f, dupesubs = 2,dedupe = t) to group reads, bbduk (parameters: qout = 33 trd = t hdist = 1 k = 27 ktrim = 'r' mink = 8 overwrite = true trimq = 10 qtrim = 'rl' threads = 10 minlength = 51 maxns = −1 minbasefrequency = 0.05 ecco = f) to remove adapter contamination, and tadpole (parameters: mode = correct, ecc = t, ecco = t) to remove sequencing error[38]. Unmatched reads were removed using bbtool's repair function. Alignment to the human genome with Bowtie2 v.2.2.3 (parameters: −very-sensitive-local) was done to remove potentially human-contaminating reads[39].

### Metagenomic assembly, bacterial and viral binning, and bin abundance quantification

We assembled all samples with MetaSPAdes v.3.14.3 (−assembler-only)[40]. Assembly quality was gauged using MetaQUAST (v.5.0.2)[41]. We binned contigs into bacterial metagenome-assembled genomes on a sample-by-sample basis using MetaBAT2 v.2.12.1 (parameters: −minContig 1500)[42]. Depth files were generated with MetaBAT2's built-in 'jgi_summarize_bam_contig_depths' function. Alignments

used in the binning process were created with Bowtie2 v.2.2.3 (parameters: −very-sensitive-local) and formatted into index bamfiles with samtools v.1.0.

Genome bin quality was checked using the 'lineage' workflow of CheckM (v.1.2)[43]. Medium and high-quality bins were dereplicated using deRep v.3.2.2 (parameters: -p 15 -comp 50 -pa 0.9 -sa 0.95 -nc 0.30 -cm larger). The resulting database of non-redundant bins was formatted as an xtree database (parameters: xtree BUILD k 29 comp 2), and sample-by-sample alignments and relative abundances were completed with the same approach as before. Bins were assigned taxonomic annotations with GTDB-tK (v.2.1.1)[44].

### Identification and taxonomic annotation of assembled viral contigs

To identify putative viral contigs, we used CheckV (v.0.8.1)[45]. For downstream viral abundance quantification, we filtered for contigs annotated as medium quality, high quality or complete. This contig database was dereplicated using BLAST and clustered at the 99% identity threshold as described above using established and published approaches (https://github.com/snayfach/MGV/tree/master/ani_cluster)[46]. The non-redundant viral contigs were formatted as an xtree database (parameters: xtree BUILD k 29 comp 0), and sample-by-sample alignments and relative abundances were computed with the same approach as before, the only difference being the coverage cut-off used to filter out viral genomes, which was lowered to 1% total and 0.05% unique due to the fact that those in question came directly from the samples analysed.

### Gene catalogue construction and functional annotation

We generated gene catalogues using an approach piloted in previous studies[47–49]. Bakta v.1.5.1 was used to call putative open reading frames (ORFs)[50]. The annotations reported in this study (for example, Fig. 5) derive directly from Bakta. We clustered predicted and translated ORFs (at 90% requisite overlap and 90% identity) into homology-based sequence clusters using MMseqs2 v.13.4511 (parameters: −easy-cluster −min-seq-id 0.9 -c 0.9)[51]. The resulting 'non-redundant' gene catalogue and its annotations were used in the functional analysis. We computed the abundance of the representative consensus sequences selected by MMseqs2 by alignment of quality-controlled reads with Diamond (v.2.0.14)[52]. We computed the total number of hits and computed gene relative abundance by dividing the number of aligned reads to a given gene by its length and then by the total number of aligned reads across all genes in a sample.

### Benchmarking short-read viral taxonomic classification against the GenBank database

To identify viral taxonomic abundance via short-read alignment, we mapped reads to a database of all complete, dereplicated (by BLAST at 99% sequence identity) GenBank viral genomes. We used the Xtree aligner for this method (see below); however, given the difficulty of assigning taxonomic ranks to viral species on the basis of alignment alone, we first benchmarked this process. We used Art[53] to generate synthetic viral communities at random abundances from 100 random viruses from the GenBank database. We then aligned (with Xtree) back to these genomes, filtered for 1% total coverage and/or 0.5% unique coverage, and compared expected read mapping vs observed read mapping. We additionally computed true/false positive rates on the basis of the proportion of taxa identified that were present in the mock community (true positive) versus those that were not (false positive) versus those that were present but not identified (false negative). Overall, we identified optimal classification at the genus level, with >98% true positive rate (that is, 98/100 taxa identified) and low false positive/negative rates (for example, <10 taxa not present in the sample identified) (Extended Data Fig. 10a,b). Species-level classification had higher false negative rates (generally arising from multimapping reads

to highly similar species) and a 60–70% true positive rate. Genus-level classification also yielded a nearly perfect correlation (>0.99 on average) between expected and observed read mappings (Extended Data Fig. 10c). As a result, while we report analyses for every taxonomic rank in the supplement, in the main text we describe only genus-level viral analysis.

### Overview of short-read taxonomic classification via alignment

In total, we used and compared seven different short-read mapping methods (MetaPhlAn4/StrainPhlAn, Xtree, Kraken2/Bracken run with four different settings, and Phanta), which together utilize five different databases that span bacterial, viral and fungal life. In addition, we identified and computed the relative abundance of non-redundant genes as well as bacterial and viral metagenome-assembled genomes. Subsequent downstream regression analyses were run on each resultant abundance table at each taxonomic rank.

Unless otherwise stated, for the figures involving taxonomic data used in the main text of this paper, we used XTree (https://github.com/GabeAl/UTree) (parameters: –redistribute). XTree is a recent update to Utree[54] containing an optimized alignment approach and increased ease of use. In brief, it is a *k*-mer-based aligner (akin to Kraken2 (ref. 55) but faster and designed for larger databases) that uses capitalist read redistribution[56] to pick the highest-likelihood mapping between a read and a given reference based on the overall support of all reads in a sample for said reference. It reports the total coverage of a given query genome, as well as total unique coverage, which refers to coverage of regions found in only one genome of an entire genome database. We computed beta diversity (Bray–Curtis) metrics for taxonomic abundances using the vegan package in R[57].

### Bacteria-specific short-read classification

For bacterial alignments, we generated an Xtree *k*-mer database (parameters: BUILD k 29 comp 0) from the Genome Taxonomy Database representative species dataset (Release 207) and aligned both metagenomic and metatranscriptomic samples. We filtered bacterial genomes for those that had at least 0.5% coverage and/or 0.25% unique coverage. Relative abundance was calculated by dividing the total reads assigned to a given genome by the total number of reads assigned to all genomes in a given sample. We additionally ran MetaPhlAn4 (ref. 58) (default settings) as an alternative approach to bacterial taxonomic classification.

### Virus-specific short-read classification

For viral GenBank alignments, we generated an Xtree database (parameters: BUILD k 17 comp 0) from all complete GenBank viral genomes. We first dereplicated these sequences with BLAST 99% identity threshold via published approaches (https://github.com/snayfach/MGV/tree/master/ani_cluster)[46,59]. We filtered for genomes with 1%/0.5% total/unique coverage. Relative abundance was calculated identically as with the bacterial samples. We additionally ran Phanta (default settings) as an alternative to this approach for viral classification[60].

### Kraken2 (multikingdom) short-read classification

As another set of methods for measuring taxonomic sample composition, we used Kraken2 and bracken, both with the default settings, to call taxa and quantify their abundances, respectively[55,61]. We used the default kraken2 reference databases, which include all NCBI listed taxa (bacteria, fungal and viral genomes) in RefSeq as of September 2022. We ran Kraken2 with four different settings: default (confidence = 0) and unmasked reads, confidence = 0 and masked reads, confidence = 0.2 and unmasked reads, and confidence = 0.2 and masked reads. In the cases where we masked reads before alignment (to filter repeats and determine whether fungal and other eukaryotic alignments were probably false positives), we used bbmask running default settings.

### Evaluation of bacterial and viral short-read classification

To evaluate our taxonomic profiling approach, we first compared the top ten genus-level classifications by body site before and after decontamination for each classifier in metagenomic and metatranscriptomic data. We observed general concordance among the various classification methods; for instance, the predominant skin genera consistently identified included *Staphylococcus*, *Cutibacterium* and *Corynebacterium*. The oral microbiome included *Streptococcus*, *Rothia* and *Fusobacterium*. Kraken2, which uses a database comprising both eukaryotic and prokaryotic organisms, identified fungi in the skin microbiome, as expected. The swabs from the Dragon capsule predominantly contained a diverse array of environmental microbes.

We compared these results at additional taxonomic ranks and with other taxonomic classifiers. For example, to discern higher specificity of the viral changes, we additionally fit species-level virus associations. While species-level viral taxonomic classification can be difficult due to high read misalignments (Extended Data Fig. 10), we wanted to determine whether we could observe a higher-resolution picture of viral activity due to spaceflight, as this effect is known to be space-associated (as opposed to bacterial skin to skin transmission, which could be a result of sharing tight quarters and not a space-specific effect).

### Sample decontamination with negative controls

We observed that many of the swabs collected, especially those from the skin sites, comprised low-biomass microbial communities; there are many documented challenges in analysing these data[62,63]. To filter environmental contamination and the kitome[64] influencing our findings, we collected and sequenced negative controls of both (1) the water that sterile swabs were dipped in before use, as well as (2) the ambient air around the sites of sample collection and processing for sequencing.

Following taxonomic classification and identification of de novo assembled microbial genes, we removed potential contaminants from samples by comparison to our negative controls. We ran the same classification approaches for each negative control sample as described in the above paragraphs. This yielded, for every taxonomy classification approach and accompanying database, a dataframe of negative controls alongside a companion dataframe of experimental data. On each of these dataframe pairs, we then used the isContaminant function (parameters: method = 'prevalence', threshold = 0.5) of the decontam package[65] to mutually high-prevalence taxa between the negative controls and experimental samples. The guidance for implementation of the decontam package, including the parameter used, was derived from the following R vignette: https://benjjneb.github.io/decontam/vignettes/decontam_intro.html. Note that we used both metagenomic and metatranscriptomic negative control samples to decontaminate all data, regardless of whether those data were themselves metagenomic or metatranscriptomic. This decision was made to increase the overall conservatism of our approach.

### MAS on bacteria, viruses and genes

Four mixed-model specifications were used for identifying microbial feature relationships with flight. Time is a variable encoded with three levels corresponding to the time of sampling relative to flight: pre-flight, mid-flight and post-flight. The reference group was the mid-flight timepoint, indicating that any regression coefficients had to be interpreted relative to flight (that is, a negative coefficient on the pre-launch timepoint implies that a feature was increased in-flight). We fit these models for all genes, viruses, and bacteria identified in our dataset by assembly, XTree (GTDB/GenBank), MetaPhlAn4, Kraken2 (all four algorithmic specifications), Phanta and gene catalogue construction. Each variable encoding a body site is binary, encoding whether a sample did or did not come from a particular region.

To search for features that were changed across the entire body, we fit overall associations, oral associations, skin associations and nasal associations:

$$\ln(\text{microbial\_feature\_abundance} + \text{minval})$$
$$\sim \beta_0 + \beta_1 \text{Time} + (1|\text{Crew.ID}) + \epsilon_i \quad (1)$$

For associations with oral changes, we used:

$$ln(\text{microbial\_feature\_abundance} + \text{minval})$$
$$\sim \beta_0 + \beta_1 \text{Time} \times \text{Oral} + (1|\text{Crew.ID}) + \epsilon_i \quad (2)$$

For associations with nasal changes, we used:

$$\ln(\text{microbial\_feature\_abundance} + \text{minval})$$
$$\sim \beta_0 + \beta_1 \text{Time} \times \text{Nasal} + (1|\text{Crew.ID}) + \epsilon_i \quad (3)$$

For identifying associations with skin swabs, we fit the following model:

$$\ln(\text{microbial\_feature\_abundance} + \text{minval})$$
$$\sim \beta_0 + \beta_1 \text{Time} \times \text{Armpit} + \beta_2 \text{Time} \times \text{ToeWeb} + \beta_3 \text{Time} \times \text{NapeOfNeck}$$
$$+ \beta_4 \text{Time} \times \text{Postauricular} + \beta_5 \text{Time} \times \text{Forehead} + \beta_6 \text{Time} \times \text{BellyButton}$$
$$+ \beta_7 \text{Time} \times \text{GlutealCrease} + \beta_8 \text{Time} \times \text{TZone} + (1|\text{Crew.ID}) + \epsilon_i$$
$$(4)$$

The β characters in each of the above equations refer to the beta coefficients on a given variable in that given regression. The ε characters refer to the regression residuals. Note that in the final equation (4), the reference groups are samples deriving from the nasal and oral microbiomes; this means that highlighted taxa will be those associated with time and skin sites as compared to the oral and nasal sites. We additionally fit these same model specifications without the random effect and compared the results in Extended Data Fig. 2. Data distributions were assumed to be normal but not tested for every single microbial feature. Individual data points for each feature are present in the online data stored at figshare[66] and with NASA GeneLab (see Data availability).

We used the lme4 (ref. 67) package to compute associations between microbial features (that is, taxa or genes) abundance and time as a function of spaceflight and body site. For all data types, we aimed to remove potential contamination before running any associations. We estimated P values on all models with the lmerTest package using its default settings[67,68]. We adjusted for false positives using Benjamini–Hochberg adjustment and used a q-value cut-off point of 0.05 to gauge significance.

### Identifying and plotting time-dependent trends in microbial features
We grouped microbial features associated with flight into six different categories. These were determined since our model contained a categorical variable encoding a sample's timing relative to flight: whether it was taken before, during or afterwards. Since the modelling reference group was 'mid-flight', the interpretation of any coefficients would be directionally oriented relative to mid-flight microbial feature abundances. As a result, we were able to categorize features on the basis of the jointly considered direction of association and significance for the 'pre-flight' and 'post-flight' levels of this variable. The below listed categories are all included in the association summaries provided on figshare[66] (see 'Data availability').

1. Transient increase in-flight—negative coefficient on the pre-flight variable level, negative coefficient on the post-flight variable, statistically significant for both
2. Transient increase in-flight (low priority)—negative coefficient on the pre-flight variable level, negative coefficient on the post-flight variable, statistically significant for at least one of the two
3. Transient decrease in-flight—positive coefficient on the pre-flight variable level, positive coefficient on the post-flight variable level, statistically significant for both

4. Transient decrease in-flight (low priority)—positive coefficient on the pre-flight variable level, positive coefficient on the post-flight variable level, statistically significant for at least one of the two
5. Potential persistent increase—negative coefficient on the pre-flight variable level, positive coefficient on the post-flight variable level, statistically significant for at least one of the two
6. Potential persistent decrease—positive coefficient on the pre-flight variable level, negative coefficient on the post-flight variable level, statistically significant for at least one of the two

We used these groups to surmise the time trends reported in the figures. It would be intractable to visualize every association of interest, so we prioritized within each category on the basis of the absolute value of beta-coefficients and adjusted P values. In Fig. 1c, we removed the 'low priority' categories (two and four above) and only looked at the top 100 most increased and decreased significant genes, by group, relative to flight. We did so to make fitting splines feasible (especially in the case of genes, which had so many associations) and filter out additional noise due to low association-size findings.

We took a similar approach for the barplots in Figs. 2–4 and Extended Data Figs. 7–9. We again filtered out the low priority associations and selected, for each body site represented in the figure (for example, oral, skin, nasal), the top $N$ with the greatest difference in absolute value of average $L_2FC$ relative to the mid-flight timepoints. In other words, we selected for microbial features with dramatic overall $L_2FC$s. We maximized $N$ on the basis of the available space in the figure in question. We note that the complete, categorized association results are available in the online data resource (see Data availability).

### Detecting microbial sharing between the crew and environment before, during and after flight
We modelled our species/strain-sharing analysis on the basis of ref. 30. Briefly, we used the –s flag in MetaPhlAn4 to generate sam files that could be fed into StrainPhlAn. We used the sample2markers.py script to generate consensus markers and extracted markers for each identified strain using extract_markers.py. We ran StrainPhlAn with the settings recommended in ref. 30 (–markers_in_n_samples 1, –samples_with_n_markers 10 –mutation_rates –phylophlan_mode accurate). We then used the tree distance files generated by StrainPhlAn to identify strain-sharing cut-offs on the basis of the prevalence of different strains (detailed tutorial: https://github.com/biobakery/MetaPhlAn/wiki/Strain-Sharing-Inference).

### Association with host immune gene subtypes
The single-cell sequencing approach and averaging of host genes to identify expression levels are documented in refs. 33,69. The resultant averaged expression levels across cell types were associated with microbial feature abundance/expression using lasso regression. We used the same log transformation approach as in the mixed effects modelling for the microbial features, and we centred and rescaled the immune expression data. In total, we computed one regression per immune cell type ($N = 8$) per relevant microbial feature, with the independent variables being all human genes ($N = 30,601$). We selected features on the basis of their grouping described above, picking only those that were increased transiently or persistently increased after flight. Due to the volume of gene-catalogue associations, we only analysed persistently increased genes. We report outcomes with non-zero coefficients in the text.

### Figure generation and additional data processing notes
The GNU parallel package was used for multiprocessing on the Linux command line[70]. We additionally used a series of separate R packages for analysis and visualization[67,68,71–76]. Figures were compiled in Adobe Illustrator.

## Statistics and reproducibility

No statistical method was used to predetermine sample size; all possible samples from all crew members ($N = 4$) were taken. No sequenced data were excluded from the analyses; however, samples were quality controlled before bioinformatic and statistical analysis to remove duplicated reads, trim adapters and low-quality bases, remove human contamination and remove potential microbial contamination (using negative controls). The experiments were not randomized. Data collection and analysis were not performed blind to the conditions of the experiments.

## Reporting summary

Further information on research design is available in the Nature Portfolio Reporting Summary linked to this article.

## Data availability

The data that support this study are available at the NASA GeneLab/NASA Open Science Data Repository with the identifiers OSD-630 (https://doi.org/10.26030/cyfk-5f38), OSD-570 (https://doi.org/10.26030/41s1-j243), OSD-572 (https://doi.org/10.26030/8v5d-xn21) and OSD-573 (https://doi.org/10.26030/x57b-4722). Additional processed datasets (gene catalogues, taxonomic and gene abundances) are available on figshare at https://figshare.com/projects/Longitudinal_multi-omics_analysis_of_host_microbiome_architecture_and_immune_responses_during_short-term_spaceflight/176043 (ref. 66). This figshare repository additionally contains figures detailing the top most abundant taxa for each alignment algorithm before and after decontamination. Select data can be visualized online through the SOMA Data Explorer: https://soma.weill.cornell.edu. The GenBank viral database used was the most recent as of 26 July 2022. The GTDB database used was the 202 release. The MetaPhlan4 database was mpa_vJan21_CHOCOPhlAnSGB_202103. The Kraken2 database contained all NCBI listed taxa (bacteria, fungal and viral genomes) in RefSeq as of 1 September 2022. The Phanta database was the most recent as of 1 August 2022. The Bakta databases were the most recent as of 18 August 2022. Source data are provided with this paper.

## Code availability

Code used to generate figures and analyses from this project is available at https://github.com/eliah-o/inspiration4-omics.

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

## Acknowledgements

We thank the WorldQuant Foundation, the Scientific Computing Unit (SCU) at WCM, NASA (NNX14AH50G, NNX17AB26G, 80NSSC22K0254, NNH18ZTT001N-FG2, 80NSSC23K0832), L. Radvinsky, K. Chudnovsky, the National Institutes of Health (R01MH117406, P01CA214274 R01CA249054, R01ES032638, R01AI151059), the LLS (MCL7001-18, LLS 9238-16, 7029-23), and the GI Research Foundation (GIRF). We also thank J. Gandara at the Microbiome Core Lab at Weill Cornell Medical College for sequencing support. J.K. thanks MOGAM Science Foundation and was supported by the Basic Science Research Program through the National Research Foundation of Korea (NRF) funded by the Ministry of Education (RS-2023-00241586). We acknowledge curation support from the NASA Open Science Data Repository/GeneLab, which is funded by the NASA Space Biology Program (Science Mission Directorate, Biological and Physical Sciences Division). We thank Boryung for support and the Global Space Healthcare Initiative and Humans in Space Program. We thank B. Kent for drawing the cartoon in Fig. 1. Figure 1 was created in part with BioRender.com.

## Author contributions

C.E.M., B.T.T. and E.G.O. conceptualized and designed the study. B.T.T. led the paper drafting, data organization and processing. All authors read and approved the paper and contributed editing, analytic recommendations, and/or assistance in responding to reviewers.

## Competing interests

B.T.T. was compensated for consulting with Seed Health and Enzymetrics Biosciences on microbiome study design. R.D. and G.A.A.-G. are employees of Seed Health. C.E.M. is a co-founder of Cosmica Biosciences. E.E.A. is a consultant for Thorne HealthTech. G.M.C. has conflicts, detailed here: https://arep.med.harvard.edu/gmc/tech.html. J.F. and M.M. are employees of Tempus Labs. J.M., A.M.B., J.Z., B.R.L., A.A., S.K. and S.L. are employees of Element Biosciences, which sequenced a subset of samples used in this study. Unless otherwise mentioned, none of the companies listed had a role in conceiving, executing, or funding the work described here.

## Additional information

**Extended data** is available for this paper at https://doi.org/10.1038/s41564-024-01635-8.

**Correspondence and requests for materials** should be addressed to Christopher E. Mason.

**Braden T. Tierney**[1,2,18], **JangKeun Kim**[1,2,18], **Eliah G. Overbey**[1,2,3,4], **Krista A. Ryon**[1], **Jonathan Foox**[1], **Maria A. Sierra**[5], **Chandrima Bhattacharya**[5], **Namita Damle**[1], **Deena Najjar**[1,6], **Jiwoon Park**[1,2], **J. Sebastian Garcia Medina**[1,2,5], **Nadia Houerbi**[1,2], **Cem Meydan**[1,2], **Jeremy Wain Hirschberg**[1], **Jake Qiu**[1], **Ashley S. Kleinman**[1], **Gabriel A. Al-Ghalith**[7], **Matthew MacKay**[5], **Evan E. Afshin**[1,2], **Raja Dhir**[7,8], **Joseph Borg**[9], **Christine Gatt**[9], **Nicholas Brereton**[10], **Benjamin P. Readhead**[11], **Semir Beyaz**[12], **Kasthuri J. Venkateswaran**[13], **Kelly Wiseman**[14], **Juan Moreno**[14], **Andrew M. Boddicker**[14], **Junhua Zhao**[14], **Bryan R. Lajoie**[14], **Ryan T. Scott**[15], **Andrew Altomare**[14], **Semyon Kruglyak**[14], **Shawn Levy**[14], **George M. Church**[16] & **Christopher E. Mason**[1,2,3,17] ✉

[1]Department of Physiology and Biophysics, Weill Cornell Medicine, New York, NY, USA. [2]The HRH Prince Alwaleed Bin Talal Bin Abdulaziz Alsaud Institute for Computational Biomedicine, Weill Cornell Medicine, New York, NY, USA. [3]BioAstra, Inc., New York, NY, USA. [4]Center for STEM, University of Austin, Austin, TX, USA. [5]Tri-Institutional Biology and Medicine program, Weill Cornell Medicine, New York, NY, USA. [6]Albert Einstein College of Medicine, Bronx, NY, USA. [7]Seed Health, Inc., Venice, CA, USA. [8]Swiss Institute of Allergy and Asthma Research (SIAF), University of Zurich, Davos, Switzerland. [9]Department of Applied Biomedical Science, Faculty of Health Sciences, University of Malta, Msida, Malta. [10]School of Biology and Environmental Science, University College Dublin, Dublin, Ireland. [11]ASU-Banner Neurodegenerative Disease Research Center, Arizona State University, Tempe, AZ, USA. [12]Cold Spring Harbor Laboratory, Cold Spring Harbor, NY, USA. [13]Jet Propulsion Laboratory, California Institute of Technology, Pasadena, CA, USA. [14]Element Biosciences, San Diego, CA, USA. [15]KBR; Space Biosciences Division, NASA Ames Research Center, Moffett Field, CA, USA. [16]Harvard Medical School and the Wyss Institute, Boston, MA, USA. [17]The WorldQuant Initiative for Quantitative Prediction, Weill Cornell Medicine, New York, NY, USA. [18]These authors contributed equally: Braden T. Tierney, JangKeun Kim. ✉e-mail: chm2042@med.cornell.edu

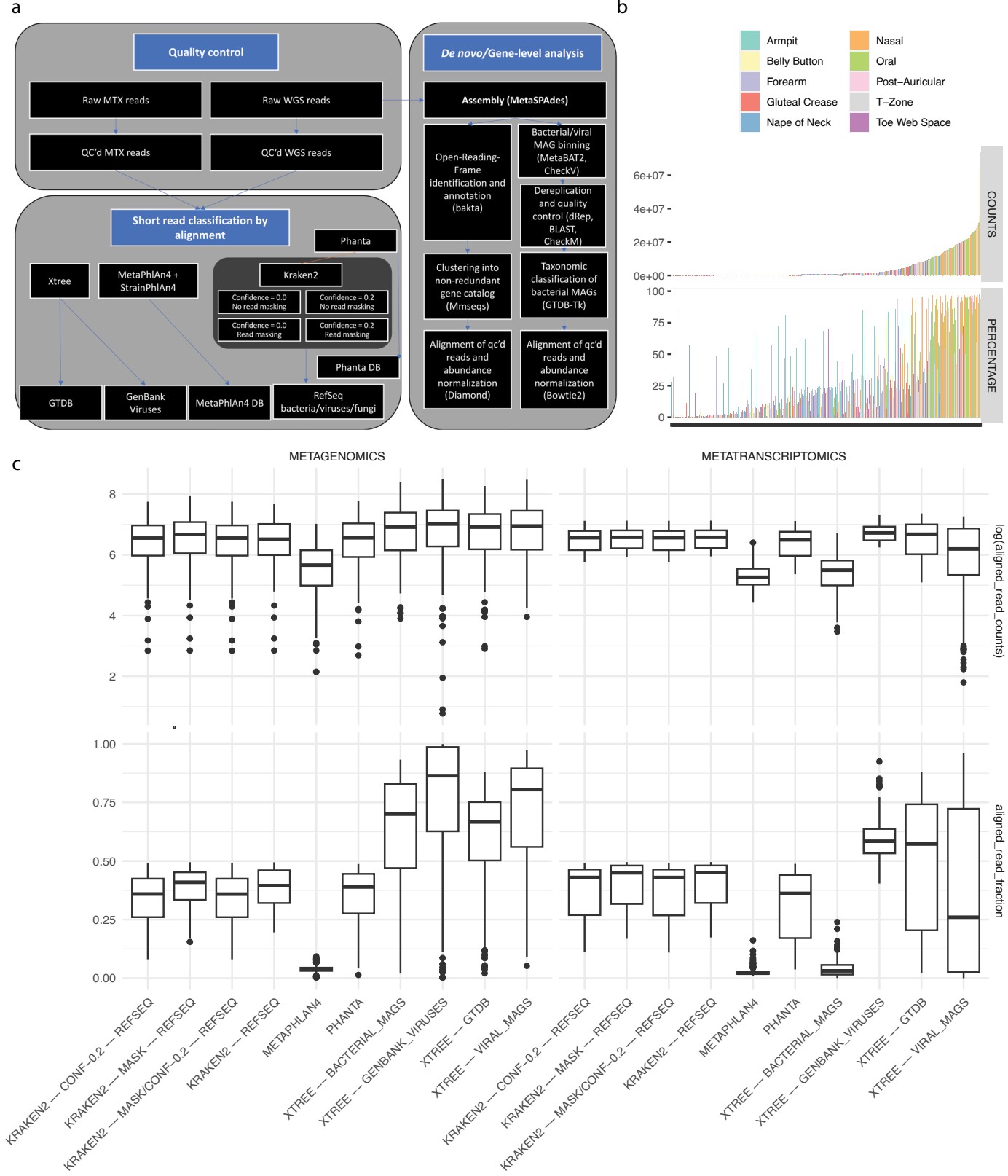

**Extended Data Fig. 1 | See next page for caption.**

**Extended Data Fig. 1 | Data processing workflow and summary statistics.**
 a) After quality-controlling reads, we executed two different, parallel, workflows to identify the microbial taxa and genes that comprised each sample. We used seven different algorithmic approaches (Xtree, MetaPhlAn4/StrainPhlAn4, Phanta, Kraken2 with multiple parameter settings) and four different databases to classify short reads into different taxonomic categories (bottom left). We also did a *de novo* assembly analysis to identify the abundance of non-redundant genes/functions as well as Metagenome-Assembled bacterial and viral genomes. We executed all regression analyses for every resultant abundance matrix across the taxonomic ranks ranging from species to phylum. **b**) Counts and percentages of reads aligning to the human reference genome. **c**) Aligned reads by taxonomic classification method. For metagenomics, N per column is 385 biologically independent samples, for metatranscriptomics, N is 365 biologically independent samples. These numbers correspond to all microbiome samples collected. Lines on box plots indicate minimum and maximum values. The median is the centerline, and the bounds of the box are the interquartile range. The whiskers extend to 1.5 times the interquartile range of the upper and lower quartiles.

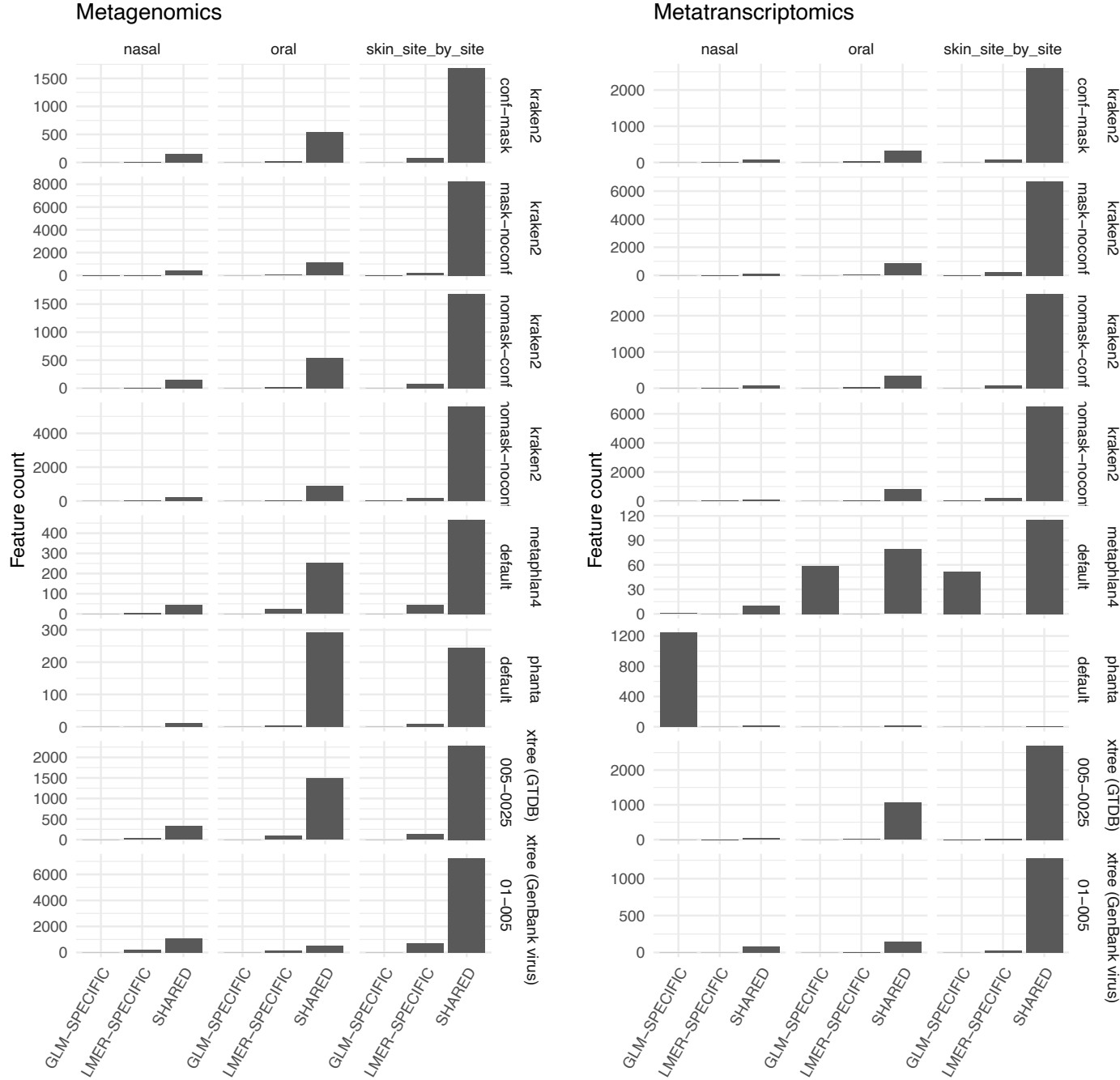

**Extended Data Fig. 2 | Null model results.** Similarity between FDR-significant associations fit with mixed versus generalized linear models (sans a random effect).

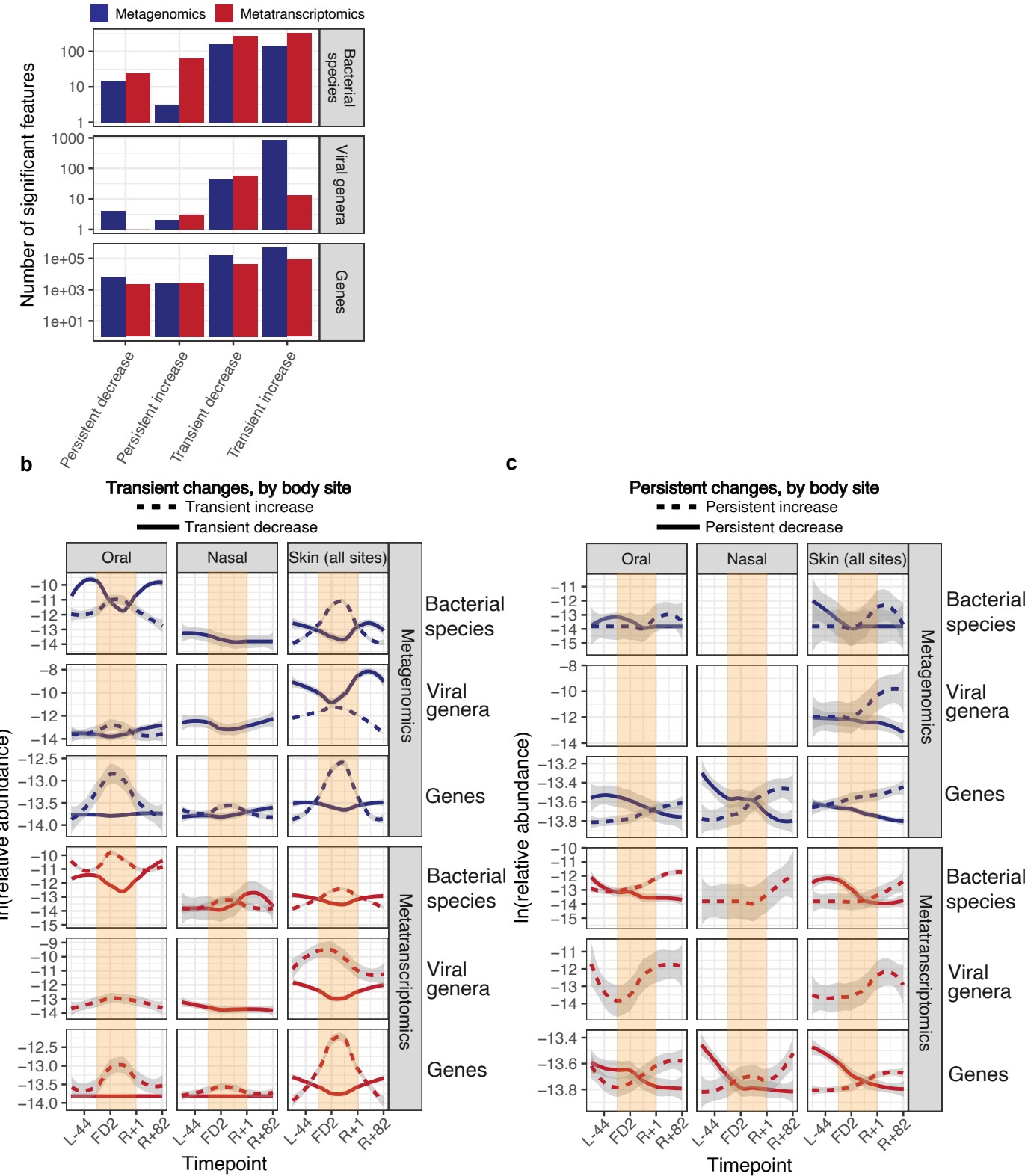

**Extended Data Fig. 3 | Supplemental Microbiome Association Study output. a)** The total number of features (bacterial species, viral genera, or genes) found to be statistically associated with either pre- or post-flight timepoints across sequencing methods. Features are grouped by the categories laid out in the *Methods* regarding the nature of their changes relative to flight **b**–**c**) The time

trajectories of persistently/transiently increased/decreased significant findings split by body site, filtering for strong [see *Methods*] associations. Plots with one or no lines had either no significant findings or none that met the filtering criteria. Gray shaded area indicates 95% confidence intervals.

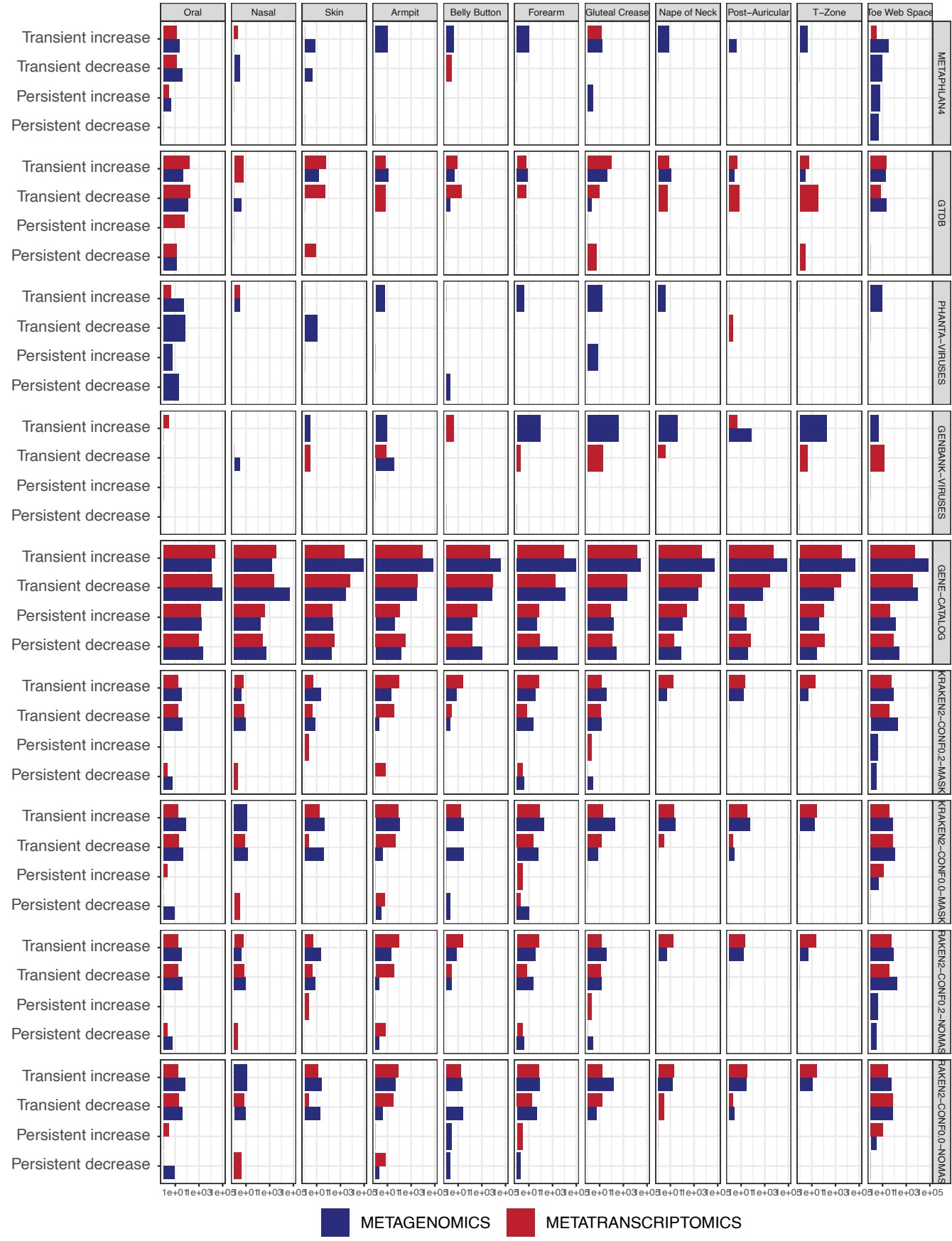

**Extended Data Fig. 4 | Regression results by specific body sites.** Regression results across short-read taxonomic classification methods.

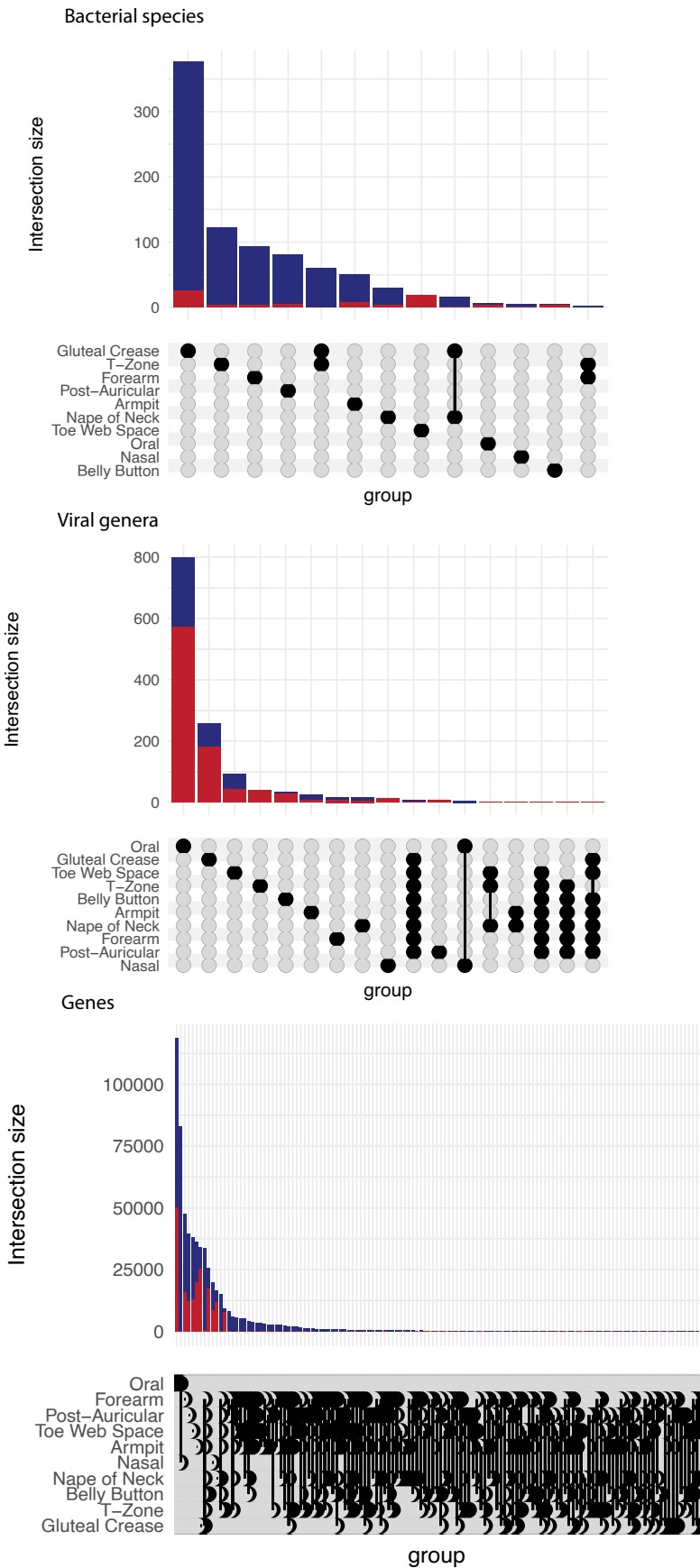

**Extended Data Fig. 5 | Similarity of regression output by body site.** Degree of overlap in the identity of significant bacterial and viral features as a function of body site and sequencing type.

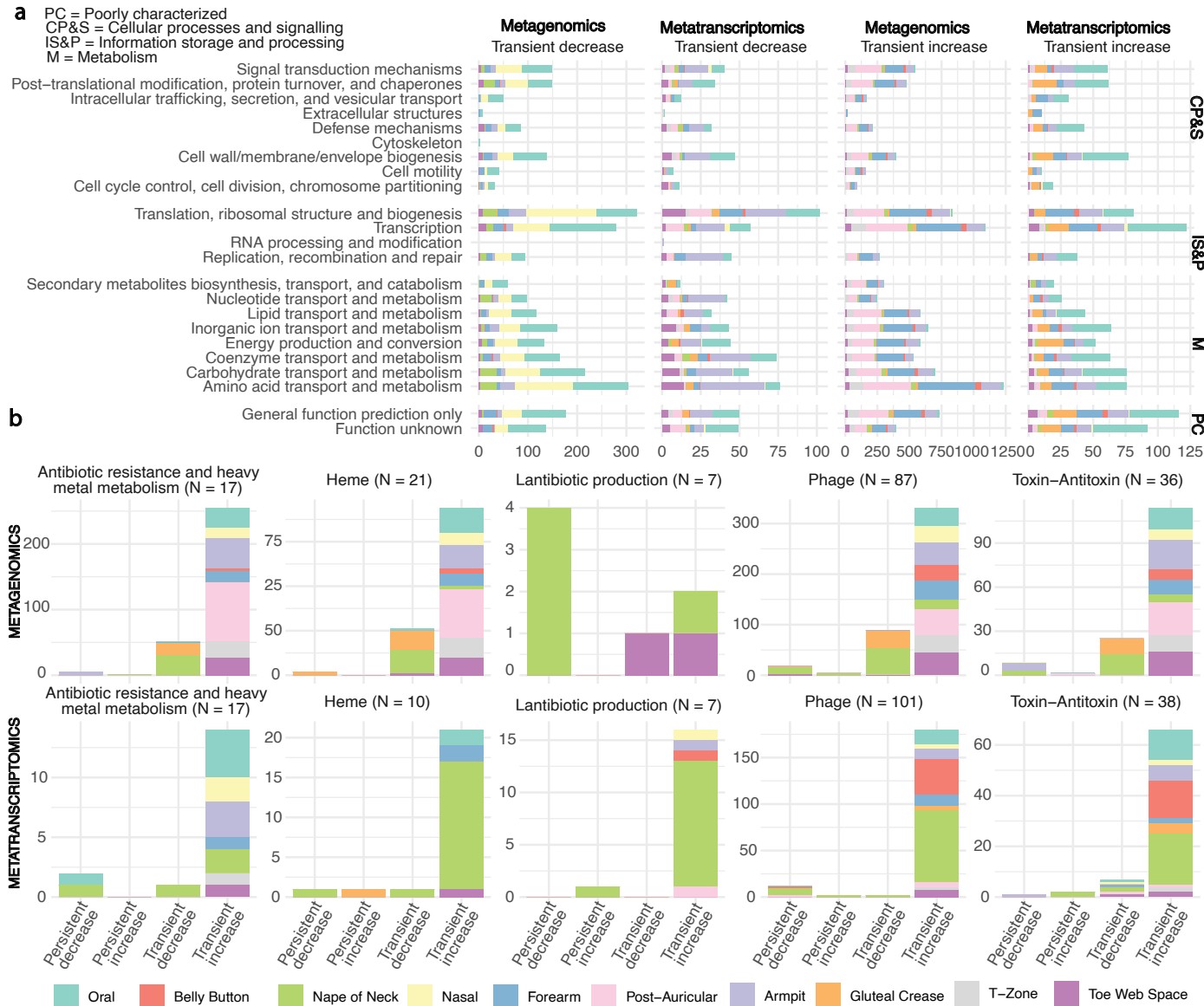

**Extended Data Fig. 6 | The functional response of the microbiome to spaceflight. a)** COG categories of all genes associated with flight. **b)** Groups of specific protein products that were associated with spaceflight. The legend in the black box is relevant for all figures where those colors appear.

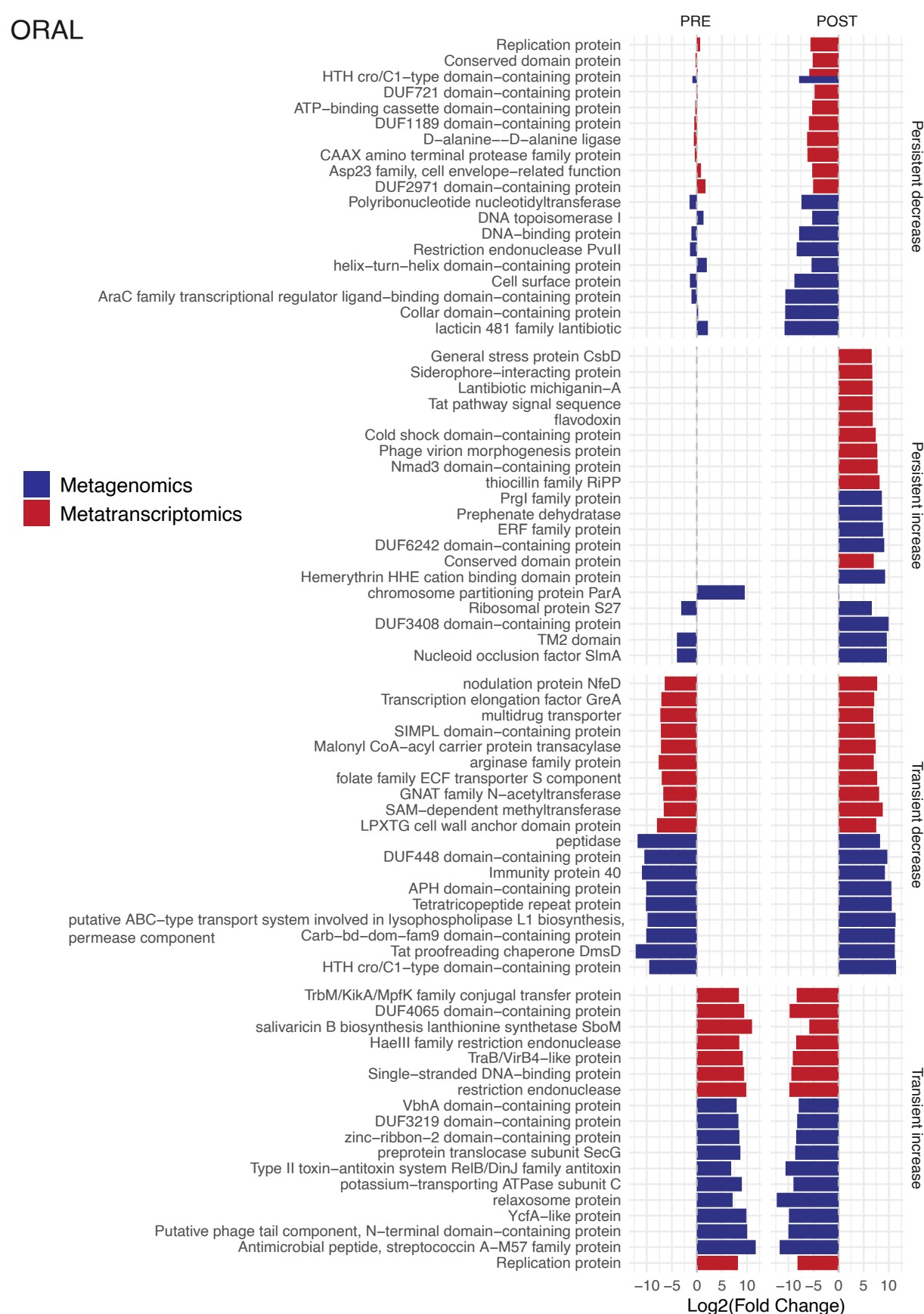

**Extended Data Fig. 7 | See next page for caption.**

**Extended Data Fig. 7 | Gene level analysis, oral microbiome.** The strongest associations between genes and flight for the oral microbiome. X-axes are the average L2FC of all pre- or post-flight timepoints compared to the average mid-flight abundances for a given taxon. Columns correspond to different association categories that are described visually by the example line plots on top of each one. Dotted, gray, horizontal lines demarcate an L2FC of zero. Plotted taxa were selected by ranking significant features in each category by L2FC and showing up to 10 at once.

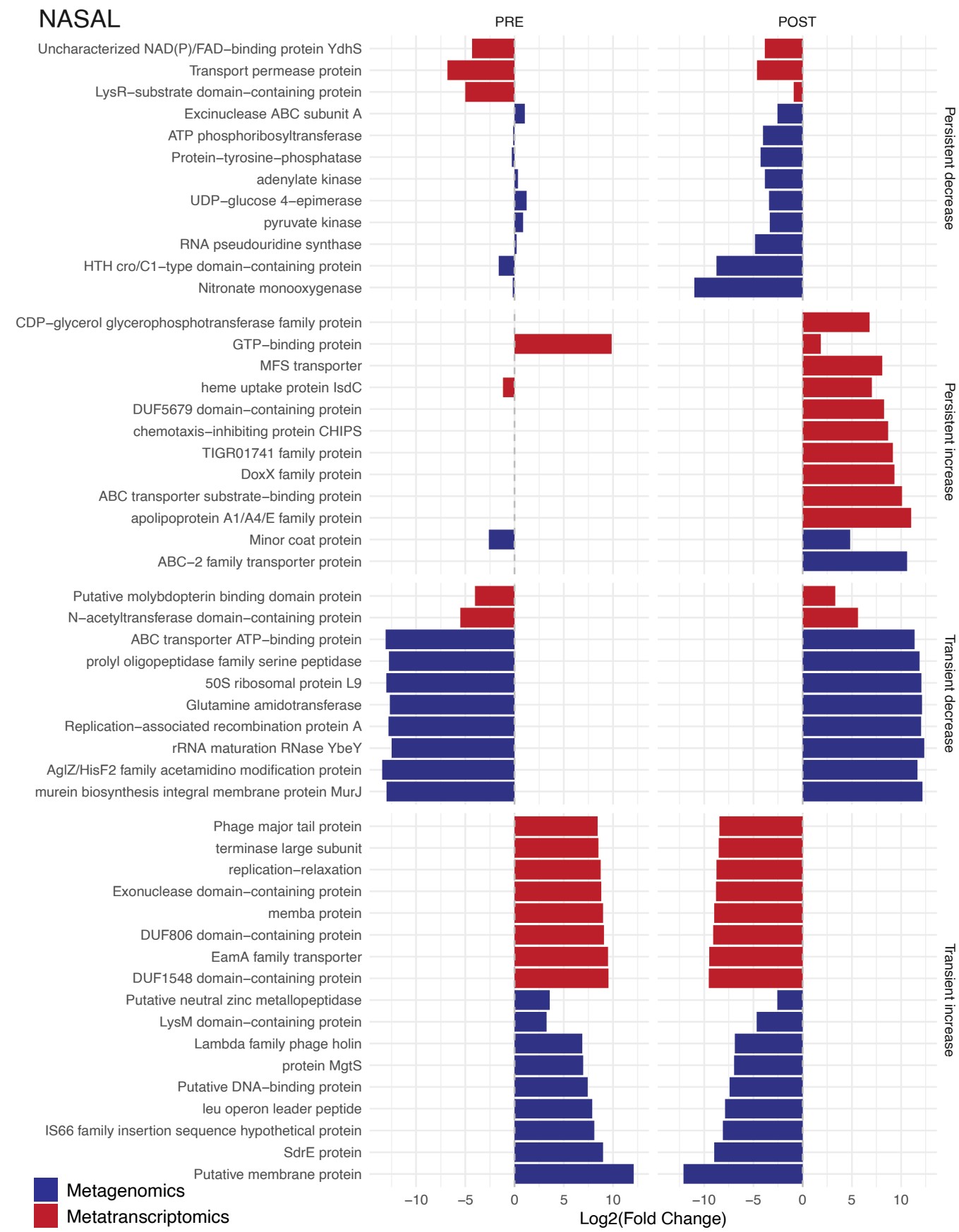

**Extended Data Fig. 8 | See next page for caption.**

**Extended Data Fig. 8 | Gene level analysis, nasal microbiome.** The strongest associations between genes and flight for the nasal microbiome. X-axes are the average L2FC of all pre- or post-flight timepoints compared to the average mid-flight abundances for a given taxon. Columns correspond to different association categories that are described visually by the example line plots on top of each one. Dotted, gray, horizontal lines demarcate an L2FC of zero. Plotted taxa were selected by ranking significant features in each category by L2FC and showing up to 10 at once.

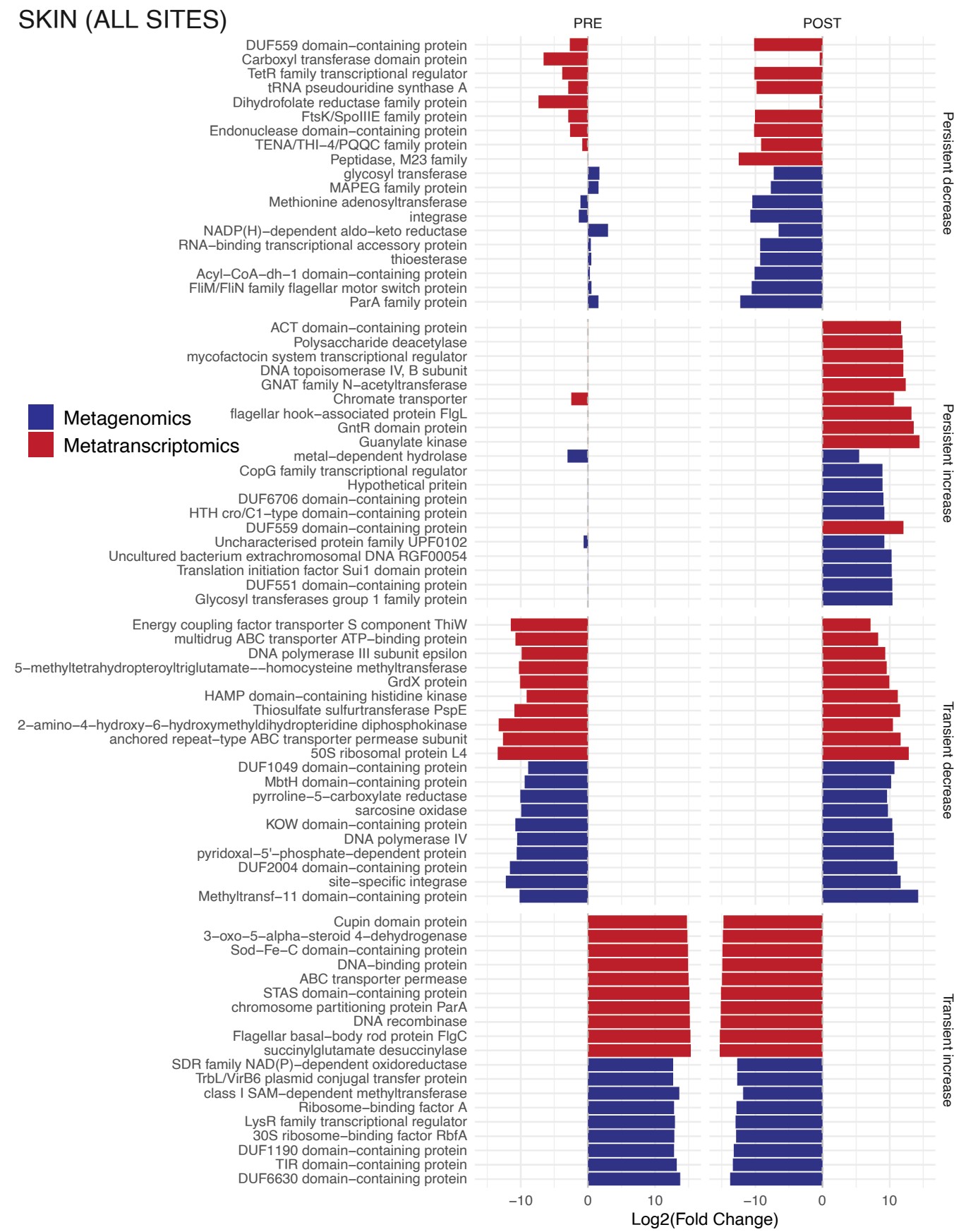

SKIN (ALL SITES)

**Extended Data Fig. 9 | See next page for caption.**

**Extended Data Fig. 9 | Gene level analysis, skin microbiome.** The strongest associations between genes and spaceflight for the skin microbiome. X-axes are the average L2FC of all pre- or post-flight timepoints compared to the average mid-flight abundances for a given taxon. Columns correspond to different association categories that are described visually by the example line plots on top of each one. Dotted, gray, horizontal lines demarcate an L2FC of zero. Plotted taxa were selected by ranking significant features in each category by L2FC and showing up to 10 at once.

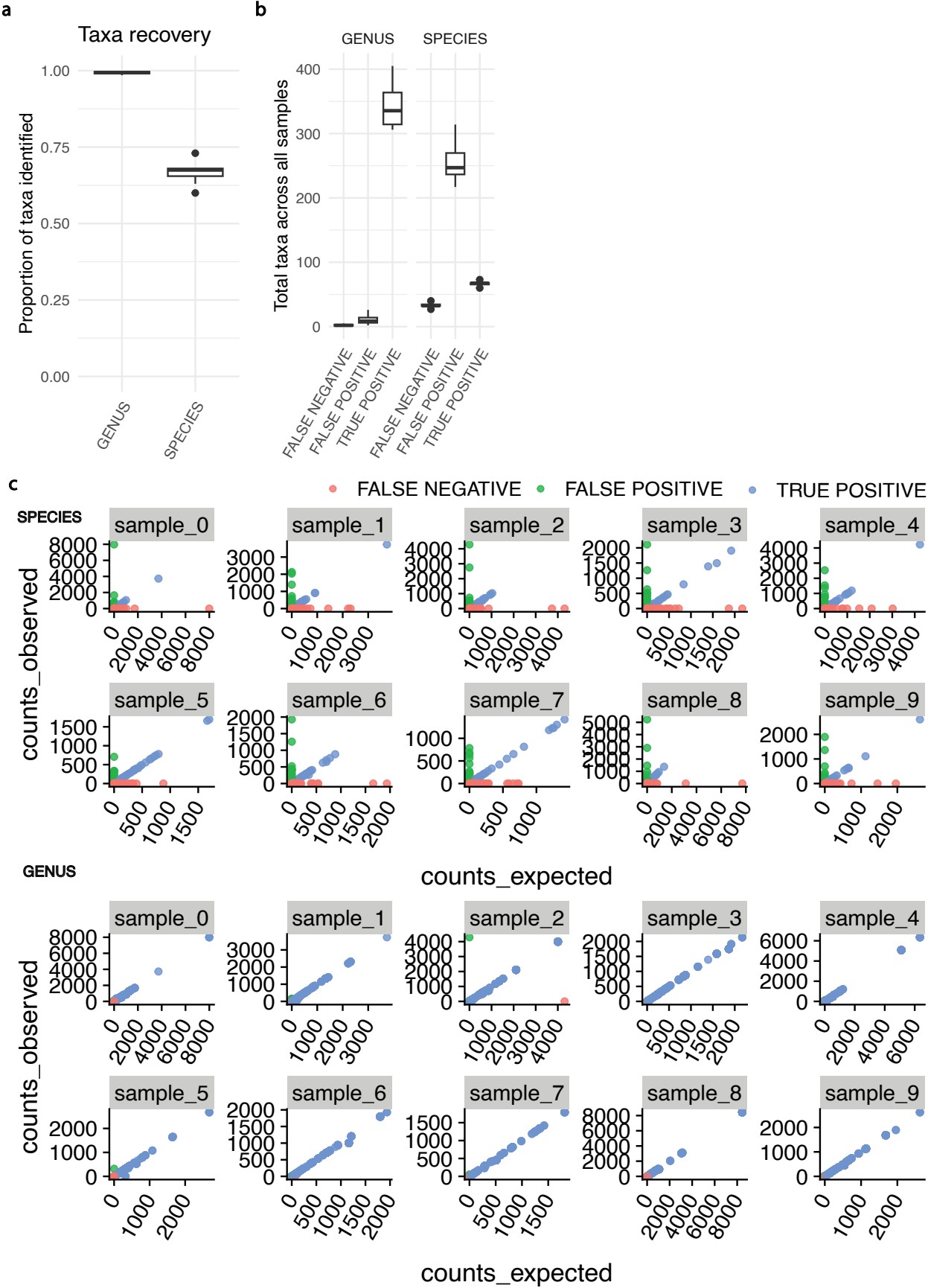

**Extended Data Fig. 10 | See next page for caption.**

**Extended Data Fig. 10 | Viral classifier benchmarking.** Benchmarking a viral classifier across taxonomic ranks. Synthetic viral communities were generated from 100 genomes at random levels of abundance (from the GenBank database used in the rest of this study). **a**) The number of recovered genomes out of 100, for 10 mock communities for the genus and species levels. N = 10 independently generated mock communities. **b**) The number of true positive (identified and present in the sample), false positive (identified but not present in the sample), and false negative (that is, not recovered) genomes for the genus and species levels. N = 10 independently generated mock communities. **c**) The correlation between observed and expected read counts for each taxon as a function of being a true positive, false positive, or false negative. Lines on box plots in A and B indicate minimum and maximum values. The median is the centerline, and the bounds of the box are the interquartile range. The whiskers extend to 1.5 times the interquartile range of the upper and lower quartiles.

# Reporting Summary

## Statistics

For all statistical analyses, confirm that the following items are present in the figure legend, table legend, main text, or Methods section.

| n/a | Confirmed | |
|---|---|---|
| ☐ | ☒ | The exact sample size (*n*) for each experimental group/condition, given as a discrete number and unit of measurement |
| ☐ | ☒ | A statement on whether measurements were taken from distinct samples or whether the same sample was measured repeatedly |
| ☐ | ☒ | The statistical test(s) used AND whether they are one- or two-sided<br>*Only common tests should be described solely by name; describe more complex techniques in the Methods section.* |
| ☐ | ☒ | A description of all covariates tested |
| ☐ | ☒ | A description of any assumptions or corrections, such as tests of normality and adjustment for multiple comparisons |
| ☐ | ☒ | A full description of the statistical parameters including central tendency (e.g. means) or other basic estimates (e.g. regression coefficient) AND variation (e.g. standard deviation) or associated estimates of uncertainty (e.g. confidence intervals) |
| ☐ | ☒ | For null hypothesis testing, the test statistic (e.g. *F*, *t*, *r*) with confidence intervals, effect sizes, degrees of freedom and *P* value noted<br>*Give P values as exact values whenever suitable.* |
| ☒ | ☐ | For Bayesian analysis, information on the choice of priors and Markov chain Monte Carlo settings |
| ☒ | ☐ | For hierarchical and complex designs, identification of the appropriate level for tests and full reporting of outcomes |
| ☒ | ☐ | Estimates of effect sizes (e.g. Cohen's *d*, Pearson's *r*), indicating how they were calculated |

*Our web collection on statistics for biologists contains articles on many of the points above.*

## Software and code

Policy information about availability of computer code

| | |
|---|---|
| Data collection | No software was used for data collection. |
| Data analysis | Code availability:<br>Code used to generate Figures and analyses from this project is available at https://github.com/eliah-o/inspiration4-omics.<br><br>Software:<br>bbtools (v38.92)XTree (v0.92i) Kraken2 (v2.l.2) bracken (v2.6.2) vegan (v2.6.2) art_illumina (v2.3.7) MetaSPAdes (v3.14.3) MetaQUAST (vS.0.2) Bowtie2 (v2.2.3) samtools (vl.0) CheckV (v0.8.1) Bakta (vl.5.1) MMseqs2 (vl3.4511) Diamond (v2.0.14) trimmomatic (v0.39) phanta (vl.1.0) metabat2 (V2.12.1) CheckM (V1.2) deRep (V3.2.2) MetaPhlan4 (v4.0.4) GTDB-tK (V2.1.1) |

For manuscripts utilizing custom algorithms or software that are central to the research but not yet described in published literature, software must be made available to editors and reviewers. We strongly encourage code deposition in a community repository (e.g. GitHub). See the Nature Portfolio guidelines for submitting code & software for further information.

## Data

Policy information about availability of data

All manuscripts must include a data availability statement. This statement should provide the following information, where applicable:
- Accession codes, unique identifiers, or web links for publicly available datasets
- A description of any restrictions on data availability
- For clinical datasets or third party data, please ensure that the statement adheres to our policy

The data that support this study is available at the NASA GeneLab/NASA Open Science Data Repository with the identifiers OSD-630 (https://doi.org/10.26030/cyfk-5f38), OSD-570 (https://doi.org/10.26030/41s1-j243), OSD-572 (https://doi.org/10.26030/x57b-4722) and OSD-573 and (https://doi.org/10.26030/x57b-4722). Additional processed datasets (gene catalogs, taxonomic and gene abundances) are available at https://figshare.com/ projects/Longitudinal_multi-omics_analysis_of_host_microbiome_architecture_and_immune_responses_during_short-term_spaceflight/176043. This Figshare repository additionally contains figures detailing the top most abundant taxa for each alignment algorithm before and after decontamination. Select data can be visualized online through the SOMA Data Explorer: https://soma.weill.cornell.edu. The GenBank viral database used was the most recent as of 2022-07-26. The GTDB database used was the 202 release. The MetaPhlan4 database was mpa_vJan21_CHOCOPhlAnSGB_202103. The Kraken2 database contained all NCBI listed taxa (bacteria, fungal, and viral genomes) in RefSeq, as of 2022-09-01. The Phanta database was the most recent as of 2022-08-01. The Bakta databases were the most recent as of 2022-08-18.

## Human research participants

Policy information about studies involving human research participants and Sex and Gender in Research.

| | |
|---|---|
| Reporting on sex and gender | Sex is reported in the manuscript. |
| Population characteristics | The crew member composition was of two races and ages ranged from 29-51. |
| Recruitment | Participants were recruited by SpaceX and mission commander Jared Isaacman; the authors did not have say in recruitment, as it was done prior to study onset. SpaceX and Mr. Isaacment recruited individuals with multiple demographic backgrounds fit for spaceflight. One won a contest, one was nominated by a US hospital, one won a raffle. Given that all individuals opted to take part in spaceflight, there be may associated self-selection biases. |
| Ethics oversight | This study was complete in accordance with appropriate ethical guidelines. All subjects were consented at an informed consent briefing (ICB) at SpaceX (Hawthorne, CA), and samples were collected and processed under the approval of the Institutional Review Board (IRB) at Weill Cornell Medicine, under Protocol 21-05023569. All crew members provided written informed consent for data and sample sharing. |

Note that full information on the approval of the study protocol must also be provided in the manuscript.

# Field-specific reporting

Please select the one below that is the best fit for your research. If you are not sure, read the appropriate sections before making your selection.

☒ Life sciences ☐ Behavioural & social sciences ☐ Ecological, evolutionary & environmental sciences

For a reference copy of the document with all sections, see nature.com/documents/nr-reporting-summary-flat.pdf

# Life sciences study design

All studies must disclose on these points even when the disclosure is negative.

| | |
|---|---|
| Sample size | The entire Inspiration4 crew was profiled, which was limited by the size of the Dragon capsule (n=4). |
| Data exclusions | No data has been excluded. |
| Replication | Replication tests were not performed; they are difficult as mission parameters could not be repeated. |
| Randomization | This is not relevant to the study as we were profiling the entire crew longitudinally (pre-flight, in-flight, and post-flight). |
| Blinding | Blinding was not possible because all subjects were astronauts in the same crew. |

# Reporting for specific materials, systems and methods

We require information from authors about some types of materials, experimental systems and methods used in many studies. Here, indicate whether each material, system or method listed is relevant to your study. If you are not sure if a list item applies to your research, read the appropriate section before selecting a response.

## Materials & experimental systems

| n/a | Involved in the study |
|-----|----------------------|
| ☒ ☐ | Antibodies |
| ☒ ☐ | Eukaryotic cell lines |
| ☒ ☐ | Palaeontology and archaeology |
| ☒ ☐ | Animals and other organisms |
| ☒ ☐ | Clinical data |
| ☒ ☐ | Dual use research of concern |

## Methods

| n/a | Involved in the study |
|-----|----------------------|
| ☒ ☐ | ChIP-seq |
| ☒ ☐ | Flow cytometry |
| ☒ ☐ | MRI-based neuroimaging |

