## [Peer Review File · Nature Microbiology]

Peer Review Information

Journal: Nature Microbiology

Manuscript Title: Longitudinal multi-omics analysis of host microbiome architecture and immune responses during short-term spaceflight

Corresponding author name(s): Dr Christopher Mason

Editorial Notes:

This manuscript has been previously reviewed at another journal. This document only contains reviewer comments, rebuttal and decision letters for versions considered at Nature Medicine. Mentions of prior referee reports have been redacted.

Reviewer Comments & Decisions:

Decision Letter, initial version:

Message: 16th October 2023

Dear Chris,

Thank you for your patience while your manuscript "Viral activation and ecological restructuring characterize a microbiome axis of spaceflight-associated immune activation" was under peer-review at Nature Microbiology. It has now been seen by 3 referees, whose expertise and comments you will find at the of this email. You will see from their comments below that while they find your work of interest, some important points are raised. We are very interested in the possibility of publishing your study in Nature Microbiology, but would like to consider your response to these concerns in the form of a revised manuscript before we make a final decision on publication.

In particular, you will see that while all of the referees agree that the manuscript is improved, they all have some remaining concerns that will need to be addressed. We feel that most of these can be addressed with text edits e.g. additional discussion, clarification, toning down claims and reworking the presentation of the data. The rest referees' reports are clear and the remaining issues should be straightforward to address.

If you have not done so already please begin to revise your manuscript so that it conforms to our Article format instructions at <http://www.nature.com/nmicrobiol/info/final-submission/>

The usual length limit for a Nature Microbiology Article is six display items (figures or tables) and 3,000 words. We have some flexibility, and can allow a revised manuscript at 3,500 words, but please consider this a firm upper limit. There is a trade-off of ~250 words per display item, so if you need more space, you could move a Figure or Table to Supplementary Information.

Some reduction could be achieved by focusing any introductory material and moving it to the start of your opening 'bold' paragraph, whose function is to outline the background to your work, describe in a sentence your new observations, and explain your main conclusions. The discussion should also be limited. Methods should be described in a separate section following the discussion, we do not place a word limit on Methods.

2Nature Microbiology titles should give a sense of the main new findings of a manuscript, and should not contain punctuation. Please keep in mind that we strongly discourage active verbs in titles, and that they should ideally fit within 90 characters each (including spaces).

Please include a data availability statement as a separate section after Methods but before references, under the heading "Data Availability". This section should inform readers about the availability of the data used to support the conclusions of your study. This information includes accession codes to public repositories (data banks for protein, DNA or RNA sequences, microarray, proteomics data etc...), references to source data published alongside the paper, unique identifiers such as URLs to data repository entries, or data set DOIs, and any other statement about data availability. At a minimum, you should include the following statement: "The data that support the findings of this study are available from the corresponding author upon request", mentioning any restrictions on availability. If DOIs are provided, we also strongly encourage including these in the Reference list (authors, title, publisher (repository name), identifier, year). For more guidance on how to write this section please see:

<http://www.nature.com/authors/policies/data/data-availability-statements-data-citations.pdf>

To improve the accessibility of your paper to readers from other research areas, please pay particular attention to the wording of the paper's opening bold paragraph, which serves both as an introduction and as a brief, non-technical summary in about 150 words. If, however, you require one or two extra sentences to explain your work clearly, please include them even if the paragraph is over-length as a result. The opening paragraph should not contain references. Because scientists from other sub-disciplines will be interested in your results and their implications, it is important to explain essential but specialised terms concisely. We suggest you show your summary paragraph to colleagues in other fields to uncover any problematic concepts.

If your paper is accepted for publication, we will edit your display items electronically so they conform to our house style and will reproduce clearly in print. If necessary, we will re-size figures to fit single or double column width. If your figures contain several parts, the parts should form a neat rectangle when assembled. Choosing the right electronic format at this stage will speed up the processing of your paper and give the best possible results in print. We would like the figures to be supplied as vector files - EPS, PDF, AI or postscript (PS) file formats (not raster or bitmap files), preferably generated with vector-graphics software (Adobe Illustrator for example). Please try to ensure that all figures are non-flattened and fully editable. All images should be at least 300 dpi resolution (when figures are scaled to approximately the size that they are to be printed at) and in RGB colour format. Please do not submit Jpeg or flattened TIFF files. Please see also 'Guidelines for Electronic Submission of Figures' at the end of this letter for further detail.

Figure legends must provide a brief description of the figure and the symbols used, within 350 words, including definitions of any error bars employed in the figures.

When submitting the revised version of your manuscript, please pay close attention to our [Digital Image Integrity Guidelines](https://www.nature.com/nature-research/editorial-policies/image-integrity). and to the following points below:

Please include a statement before the acknowledgements naming the author to whom correspondence and requests for materials should be addressed.

Finally, we require authors to include a statement of their individual contributions to the paper -- such as experimental work, project planning, data analysis, etc. -- immediately after the acknowledgements. The statement should be short, and refer to authors by their initials. For details please see the Authorship section of our joint Editorial policies at http://www.nature.com/authors/editorial_policies/authorship.html

- * include a point-by-point response to any editorial suggestions and to our referees. Please include your response to the editorial suggestions in your cover letter, and please upload your response to the referees as a separate document.
- * ensure it complies with our format requirements for Letters as set out in our guide to authors at www.nature.com/nmicrobiol/info/gta/
- * state in a cover note the length of the text, methods and legends; the number of references; number and estimated final size of figures and tables
- * resubmit electronically if possible using the link below to access your home page:

Please ensure that all correspondence is marked with your Nature Microbiology reference number in the subject line.

Nature Microbiology is committed to improving transparency in authorship. As part of our efforts in this direction, we are now requesting that all authors identified as 'corresponding author' on published papers create and link their Open Researcher and Contributor Identifier (ORCID) with their account on the Manuscript Tracking System (MTS), prior to acceptance. This applies to primary research papers only. ORCID helps the scientific community achieve unambiguous attribution of all scholarly

contributions. You can create and link your ORCID from the home page of the MTS by clicking on 'Modify my Springer Nature account'. For more information please visit please visit www.springernature.com/orcid.

We hope to receive your revised paper within three weeks. If you cannot send it within this time, please let us know.

Yours sincerely,

Reviewer Expertise:

Referee #1: human microbiome, virome
Referee #2: skin microbiome, genomics
Referee #3: microbiome, statistics

Reviewers Comments:

Reviewer #1 (Remarks to the Author):

"Viral activation..." by Tierney presents a study of human and environmental microbe dynamics during space flight. The authors analyze samples from four astronauts during a three day flight, time points before and after, and also samples from their capsule. They analyze 750 samples from ten body sites and controls. Single cell RNA-seq was used to characterize activity of the astronaut immune system. The authors found that microbial diversity increased mid-flight, particularly viruses, and dropped after flight. A minority of the changes persisted after return to Earth. The transfer of microbial species from crew members to each other and their environment were tracked. The paper is limited by sample size and the descriptive nature, but the topic is fascinating and of great interest to a wide audience.

Comments:

What negative controls were applied to the metatranscriptomics? I'm wondering how secure the detections of eukaryotic-cell viruses were in these data. It would be helpful to comment on this further.

The work is exciting, but the paper was a bit difficult to read. It presented excessive details from the analysis without being clear on why these details were important. The abstract did succeed in distilling out some of the most important points. I would suggest editing the paper further for brevity to highlight the most pertinent points.

The authors mention that "The total contents of this database are available in the NASA Open Science Data Repository (OSD-572, OSD-573)(Overbey et. al [under review]). I searched and was unable to find the data supporting this study. The authors will need to clarify whether data will be freely available, provide straightforward guidance to finding it, and clarify whether this will be the case at

the time of publication.

There were multiple typos in the text. Please edit carefully. Some were in the following lines: 77, 182, 257, 389.

Reviewer #2 (Remarks to the Author):

The writing of this manuscript has improved. The rationale for the experiments and limitations are now explained. Negative controls for skin now much better explained. However, what about the wipes that people were using to bathe themselves while in space? maybe the wipes contained microbial DNA in their solution that people are then applying to their skin. Or maybe the cleaning wipes that astronauts used caused dead DNA to be trapped on their skin. trapping of dead DNA with hospital-associated 'bath-in-bed' chlorhexidine wipes has been shown (PMID: 29753031) by skin microbiome researchers. This could explain fig 3 and fig 4B.

The discussion could be compressed a lot.

Overall, I think the microbiome data for this manuscript could be compressed from 5 figures to 2 or 3 and show the most important data that is truly significant. While the abstract states many findings, the conclusions in the paper include

'The overall lack of bacterial metagenomic signal in the skin was interesting, 367 as it indicated that strains acquired during flight that displayed altered relative abundance but 368 limited transcriptional changes did not correlate to measurable host immune response. In other 369 words, there was limited evidence that strain-sharing drove an altered immune state in humans.'

Many of the strongest bacterial skin microbiome alterations (Fig 3) were predominantly 230 metagenomic, as opposed to metatranscriptomic. We hypothesized that this may indicate the 231 acquisition of new but non-transcriptionally active species from the surrounding environment. this statement is oblique. Are the authors hypothesizing that that these microbes are dead? If so this is less of a major conclusion.

Most viral activity was transiently increased; in other words, 249 even more dramatically than in the bacterial data, relatively speaking, viral abundances reset to 250 baseline almost immediately after flight (Fig 4B).

Methods established for determining 'viral presence' in Fig 4B still has me wondering if any of this is real. Authors state that they did a benchmarking but that was with viruses from GenBank. In this case they have random shotgun reads that may or may not be weakly related viruses? Filtered for 1% total coverage or 0.5% seems extremely low. If this is just reads piling up in one location of a sequence deposited in RefSeq I don't think they can call this 'present' even at the genus level. A lot of these are odd named that seem like best match but might not hold up to scrutiny; e.g. sewage-associated circular DNA virus-20 or lake Sarah-associated circular virus-23, Gen_unclassified

Gokushovirinae (which is shown as transient decrease and transient increase on skin)? This data seemed like very small changes were being reported and of viruses that have not been associated with skin. I also don't understand how metatranscriptomics yields all transient decrease and metagenomics all yields transient increase – sometimes of the same or similar genus. Which of these are DNA versus RNA viruses?

StrainPhlAn and MetaPhlAn are designed for gut microbiome studies as authors acknowledge and yet they continue to use these marker gene based methods for skin data, undermining confidence in Figure 5. Their conclusions in text are "Overall (Fig 5B), we found that individuals appeared to acquire strains from the capsule by the 334 second mid-flight sampling point (day 3). During the L-92 timepoint, there was minimal transfer 335 between the training capsule and the astronauts." And yet their conclusion in abstract is: 'We also used 48 strain-level tracking to measure the potential propagation of microbial species from the crew 49 members to each other and the environment, identifying microbes that were prone to seed the 50 capsule surface and move between the crew.'

Methods: were the metatranscriptomics samples depleted of bacterial rRNA? Only described that human/mouse/rat [18S, 5S, 23S] was performed. Wondering if bacterial 16S rRNA also depleted?

Reviewer #3 (Remarks to the Author):

In this revision, the authors have responded to my comments on their previous draft. This is a much-improved manuscript. However, I still have some concerns listed below.

1. In response to my previous comment, the authors included pre-flight (using a different capsule) and mid-flight microbial compositions of the flight of interest. The authors claim on page 45 of their rebuttal that "... overall the metagenomic community is relatively similar on Earth vs. in-flight". I find this a bit surprising. Several people are in the space capsule in close proximity to each other. Hence, if the microbial composition can change within each subject over the three day period, then why did it not change in the capsule environment. In fact, from the relative abundance plots provided on page 45 of the rebuttal document it does appear that the capsule environment has also changed. Of course with the authors that changes in the capsule environment from earth to space is difficult to interpret because they are not the same capsule that were measured at the two time points.

2. Again, going back to my comment on the earlier draft, when microbes do not enter the capsule in flight, then how the changes in microbiome within and between subjects be explained as due to space travel? The changes in microbial compositions could be purely because several people are inside a capsule over three days. An ideal control would be if the same subjects were housed in the same capsule for three days on earth. Does the "pre-launch" data refer to this control group? I probably missed it if that is the case. Otherwise, I find the data of this project difficult to interpret.

3. In response to my comment regarding correlation analyses, the authors note on page 46 that they combined data from all sites to construct correlations. Does it mean they treated all sites as one organ for computing correlations? Since these are relative abundance data, the standard concept of Pearson correlation is not appropriate. Did they use SPARCC or some of the other recent methods? From the heatmaps, I am puzzled that all correlations among the bacterial families are all positive. Why is that? Is it an artifact of the correlation tool used? Or is there a real biology?

7

4. It is true that linear models have a history in microbiome data analysis, but recently developed algorithms such as LinDA would be better alternatives to deal with compositionality in the microbiome data. There is a risk of high false discovery rates when compositionality is ignored.

Author Rebuttal to Initial comments

Reviewers Comments:

Reviewer #1 (Remarks to the Author):

“Viral activation...” by Tierney presents a study of human and environmental microbe dynamics during space flight. The authors analyze samples from four astronauts during a three day flight, time points before and after, and also samples from their capsule. They analyze 750 samples from ten body sites and controls. Single cell RNA-seq was used to characterize activity of the astronaut immune system. The authors found that microbial diversity increased mid-flight, particularly viruses, and dropped after flight. A minority of the changes persisted after return to Earth. The transfer of microbial species from crew members to each other and their environment were tracked. The paper is limited by sample size and the descriptive nature, but the topic is fascinating and of great interest to a wide audience.

Comments:

1. What negative controls were applied to the metatranscriptomics? I’m wondering how secure the detections of eukaryotic-cell viruses were in these data. It would be helpful to comment on this further.

We executed metatranscriptomic and metagenomic sequencing on the in-flight and ground negative controls. In total, 30 controls were sequenced for metatranscriptomic samples. So, for the metatranscriptomic sequencing, we used metatranscriptomic control Isohelix swabs, which were included on every plate of samples. The updated data in the first revision included a second round of negative controls, added in accordance with the Reviewer’s requests. Of note, these were only metagenomic sequencing controls (blanks), so they were used just for the DNA swabs.

We agree that, fundamentally, decontamination via negative controls in metatranscriptomic microbiome data is challenging, especially as it relates to viral identification. For non-RNA life, algorithms are essentially filtering based on the expressed genes, which is not necessarily ideal. Even for RNA viruses, it is likely that we are removing some true-positive viruses and including false-positive ones, but our thresholds used for data processing should minimize bias. Also, we include multiple metrics and thresholds in the tables, which enables readers and researchers to select varying levels of confidence for the results they would want to use for follow-up studies.

As other Reviewers have commented, viral detection is extremely difficult and a nascent field, especially in low abundance short-read sequencing data – we have now attempted to address this in greater detail in the Supplementary Discussion, reproduced at the bottom of this document.

We additionally, as part of our revisions and efforts to hone the text, have reduced the focus on the virome. We still have one main panel showing the diversity of viral life enriched in-flight, but we no longer discuss detailed strain names outside of phyla (which are easier to annotate and of higher confidence than finer resolution clades).

2. The work is exciting, but the paper was a bit difficult to read. It presented excessive details from the analysis without being clear on why these details were important. The abstract did succeed in distilling out some of the most important points. I would suggest editing the paper further for brevity to highlight the most pertinent points.

We have continued to refine the text in the manuscript, removing nearly 2,000 words from the main text and reworking the figures for simplicity (moving one entirely to the supplement). Most notably, we have added a Supplementary Discussion (reproduced in an Appendix at the bottom of this document) section, where we can address many of the specifics of the analysis that are

important but not necessarily critical for a casual reader. This includes, for example, the caveats regarding viral detection in the above point.

3. The authors mention that “The total contents of this database are available in the NASA Open Science Data Repository (OSD-572, OSD-573)(Overbey et. al [under review]). I searched and was unable to find the data supporting this study. The authors will need to clarify whether data will be freely available, provide straightforward guidance to finding it, and clarify whether this will be the case at the time of publication.

For the last round of revisions, we provided links to where the data was being curated by NASA’s GeneLab Repository. This process is intensive and thorough – it requires uploading all files for GeneLab processing, QC, annotation, and confirmation for all ~750 sequencing files from this manuscript, but this process is finally completed. All data will be released at the time of publication, and the preview links from the NASA GeneLab repository are available for the reviewers here:

https://osdr.nasa.gov/bio/repo/data/studies/OSD-572/preview/CuhxWdYAPRX4QJRLd74yvdXjio_n2OxrXm

<https://osdr.nasa.gov/bio/repo/data/studies/OSD-573/preview/Ze2eyzuoTvJ7CI0887nweUeBe9GqYTZN>

<https://osdr.nasa.gov/bio/repo/data/studies/OSD-630/preview/BilzXsJxraKrE9sYDCsOh-MLL3yPppSC>

4. There were multiple typos in the text. Please edit carefully. Some were in the following lines: 77, 182, 257, 389.

We have addressed these – and others – as part of our editing process. We thank the Reviewer.

Reviewer #2 (Remarks to the Author):

1. The writing of this manuscript has improved. The rationale for the experiments and limitations are now explained. Negative controls for skin now much better explained. However, what about the wipes that people were using to bathe themselves while in space? maybe the wipes contained microbial DNA in their solution that people are then applying to their skin. Or maybe the cleaning wipes that astronauts used caused dead DNA to be trapped on their skin. trapping of dead DNA with hospital-associated ‘bath-in-bed’ chlorhexidine wipes has been shown (PMID: 29753031) by skin microbiome researchers. This could explain fig 3 and fig 4B.

We are glad the Reviewer is pleased with our changes – thank you. We agree that, of course, wipes used by individuals to bathe themselves in space could be a confounder.

However, we reached out to the Commander of the Inspiration4 mission, who confirmed that “any wet wipes used by the crew were not reused, nor shared,” which should limit any influence of this confounding variable. We have also noted this in the methods section for the paper, and we note the potential confounding effect of cleaning methods in the Supplementary Discussion.

In the Methods we write:

“We note that some crew members were using wet wipes [UPC: 036000317985] to bathe themselves in-flight in between swabbing; however not every crew member did so, and SpaceX did not require this to be a consistent protocol among the crew. Wet wipes used by the crew were not reused, nor shared, which should limit any influence of this confounding variable.”

And in the Discussion:

“They [sic future studies] additionally should encourage more detailed reporting on diet and cleaning methods (e.g., wet wipes) to adjust for potential confounders introducing foreign microbial DNA into the host.”

Related to consideration of the manuscript cited, we also note that it indicates “personalized” responses to wet wipe treatment. The responses we observed in our regression were, overall, shared between individuals. Also, we feel that our use of coverage thresholds and a variety of algorithmic approaches likely reduces the probability that merely lysed cells/contaminating DNA drove the majority of the observed effect.

Also, we now address the “cleaning method” as a potential confounder as well as our thoughts on it in the Supplementary Discussion section (reproduced at the bottom of this document) on drawbacks and also include the update in the methods section.

2. The discussion could be compressed a lot.

We have now trimmed down the discussion, moving some of the excessive detail to a Supplementary Discussion section where it is still accessible but less distracting.

3. Overall, I think the microbiome data for this manuscript could be compressed from 5 figures to 2 or 3 and show the most important data that is truly significant. While the abstract states many findings, the conclusions in the paper include

“The overall lack of bacterial metagenomic signal in the skin was interesting, as it indicated that strains acquired during flight that displayed altered relative abundance but limited transcriptional changes did not correlate to measurable host immune response. In other words, there was limited evidence that strain-sharing drove an altered immune state in humans.”

“Many of the strongest bacterial skin microbiome alterations (Fig 3) were predominantly metagenomic, as opposed to metatranscriptomic. We hypothesized that this may indicate the acquisition of new but non-transcriptionally active species from the surrounding environment.” this statement is oblique. Are the authors hypothesizing that that these microbes are dead? If so this is less of a major conclusion.

“Most viral activity was transiently increased; in other words, 249 even more dramatically than in the bacterial data, relatively speaking, viral abundances reset to 250 baseline almost immediately after flight (Fig 4B).”

We agree that our paper was a bit loquacious – however, we do feel it warrants more than 2-3

nature portfolio

figures, in large part to highlight the scope of the dataset, which is the largest and most diverse of its kind (it is larger than all metagenomic samples taken in space combined, as indexed in NASA's GeneLab database). To address this point, we have taken the following steps:

- 1) First, we have reduced the claims in the abstract. The Reviewer accurately calls out some of the most interesting results here, and we now focus on those.
- 2) We clarify that we are not hypothesizing these microbes are dead, just not highly transcriptionally active to the point where it could be detected by an association study. We note that many of these organisms are filtered on a coverage threshold. We are likely filtering out some true positive shifts in lieu of emphasizing high-effect size taxa.

That “filter” provided by regression means that any more minimal change will likely not be detected when compared to, say, the more dramatic metagenomic shifts on the skin.

We feel that the discordance between metagenomics and metatranscriptomics is actually one of the most interesting findings in this study, because it indicates that DNA presence does not correlate to dramatic changes in transcriptional activity. In other words, all the studies that rely on metagenomics alone, in general, may actually be missing a large portion of the picture and unable to target truly causal host-microbe relationships, which we think is an exciting result.

- 3) We have reduced focus on the viral component of our findings, relegating certain panels to the supplement and indicating that, in low abundance data, viral classification is difficult. We additionally have moved the gene catalog panels entirely to the supplement.

In total, our paper now has 5 simplified figures, and we have updated the abstract to more accurately reflect the key points highlighted by the Reviewer.

4. Methods established for determining ‘viral presence’ in Fig 4B still has me wondering if any of this is real. Authors state that they did a benchmarking but that was with viruses from GenBank. In this case they have random shotgun reads that may or may not be weakly related viruses? Filtered for 1% total coverage or 0.5% seems extremely low. If this is just reads piling up in one location of a sequence deposited in RefSeq I don’t think they can call this ‘present’ even at the genus level. A lot of these are odd named that seem like best match but might not hold up to scrutiny; e.g. sewage-associated circular DNA virus-20 or lake Sarah-associated circular virus-23, Gen_unclassified Gokushovirinae (which is shown as transient decrease and transient increase on skin)? This data seemed like very small changes were being reported and of viruses that have not been associated with skin. I also don’t understand how metatranscriptomics yields all transient decrease and metagenomics all yields transient increase – sometimes of the same or similar genus. Which of these are DNA versus RNA viruses?

In short, we agree that calling viruses is extremely difficult. The virome is not well explored. Some spurious alignments are likely false positives; others could be incomplete databases where the most related organism in the database has an unusual name.

To account for challenges in viral detection, we have done the following:

- 1) As in our first round of revisions, we worked at the genus level for higher confidence calls of taxa.
- 2) Benchmarked our approaches and used a diverse set of methods (note Supp Fig 18, where we show high accuracy at the genus level on the database we’re using in calling viruses).
- 3) Extensively discuss the difficulty of viral classification in the results and the Supplementary Discussion (reproduced below).

- 4) Reduce the emphasis on viral analysis in the manuscript, such that unusual taxa are not highlighted, focusing instead on trends in viral classes (e.g., eukaryotic vs prokaryotic) and more on the high-confidence changes.

The transient shift in viruses compared to bacteria is aligned with what we know about viral activation in space. Spaceflight changes viral communities and causes, in some cases, reactivation of latent infections (e.g., herpes viruses). This effect resets upon returning to Earth.

Overall, we claim to be using state-of-the-art methods for viral classification, and now with the reduced emphasis on viral shifts and the stated caveats in the Supplementary Discussion, we hope that our analysis is acceptable for publication.

5. StrainPhlAn and MetaPhlAn are designed for gut microbiome studies as authors acknowledge and yet they continue to use these marker gene based methods for skin data, undermining confidence in Figure 5. Their conclusions in text are “Overall (Fig 5B), we found that individuals appeared to acquire strains from the capsule by the 334 second mid-flight sampling point (day 3). During the L-92 timepoint, there was minimal transfer 335 between the training capsule and the astronauts.” And yet their conclusion in abstract is: ‘We also used 48 strain-level tracking to measure the potential propagation of microbial species from the crew 49 members to each other and the environment, identifying microbes that were prone to seed the 50 capsule surface and move between the crew.’

Thank you for this comment. We have fixed the discordance between the abstract and the text. Regarding marker gene-based approaches, we note that many methods used (e.g., Xtree) are not marker-gene based, which will help give orthogonal methods for the species measurements. In accordance with Reviewer requests in the first revision, we added MetaPhlAn4 and StranPhlAn, as they are established tools, but indeed, they were first based on stool samples.

Fortunately, the recent updates to MetaPhlAn and StranPhlAn incorporate marker genes from more than just the gut microbiome. Specifically, they use databases requested by the Reviewers (e.g., Segata and Finn labs) in the initial round of review, and they also incorporate MAGs/genomes from the skin and oral niches. Therefore, as of their recent release, the developers of these tools have expressed confidence that their updated tools are now appropriate for more than just the gut microbiome. Specifically, the recent MetaPhlAn4 paper indicated that its default database includes marker genes from more than just the gut microbiome:

“MetaPhlAn4 application to human and nonhuman metagenomes. To measure the increase of the fraction of classified reads when compared with MetaPhlAn3, we profiled 24,515 samples from 145 datasets spanning different human body sites (airways, gastrointestinal tract, oral, skin, and urogenital tract) and lifestyles, animal hosts (non-human primates, mice and ruminants) and other non-host-associated environments (soil, fresh water, and ocean).”

Indeed, the paper on which we modeled our strain-sharing approach was used for the oral microbiome (<https://www.nature.com/articles/s41586-022-05620-1>). Given these recent developments, as well as the range of other tools with which we have also processed these i4 data, we believe these results will be useful, relevant, and robust for the field.

6. Methods: were the metatranscriptomics samples depleted of bacterial rRNA? Only described that human/mouse/rat [18S, 5S, 23S] was performed. Wondering if bacterial 16S rRNA also depleted?

We did not deplete the bacterial 16S rRNA, but used the human/mouse/rat rRNA ribo-depletion kit from NEB. This method had been previously well-characterized and successful for bacterial, viral, and fungi metatranscriptomics analysis from clinical samples, and we have cited these manuscripts in our updated paper to help guide readers for more details:

<https://www.nature.com/articles/s41467-021-21361-7>

<https://genome.cshlp.org/content/early/2021/02/18/gr.268961.120.full.pdf>

Reviewer #3 (Remarks to the Author):

In this revision, the authors have responded to my comments on their previous draft. This is a much-improved manuscript. However, I still have some concerns listed below.1. In response to my previous comment, the authors included pre-flight (using a different capsule) and mid-flight microbial compositions of the flight of interest. The authors claim on page 45 of their rebuttal that "... overall the metagenomic community is relatively similar on Earth vs. in-flight". I find this a bit surprising. Several people are in the space capsule in close proximity to each other. Hence, if the microbial composition can change within each subject over the three day period, then why did it not change in the capsule environment. In fact, from the relative abundance plots provided on page 45 of the rebuttal document it does appear that the capsule environment has also changed. Of course with the authors that changes in the capsule environment from earth to space is difficult to interpret because they are not the same capsule that were measured at the two time points.

This is an important comment. Indeed, as the capsule environment is not identical in-flight as on Earth, we have remedied the relevant comment in the text to indicate the changes. We feel that it was not worth heavily focusing on them, due to our choice not to carry out statistical analysis on the capsule samples. This was, as the reviewer indicated, because the pre-flight and mid-flight capsules were different.

2. Again, going back to my comment on the earlier draft, when microbes do not enter the capsule in flight, then how the changes in microbiome within and between subjects be explained as due to space travel? The changes in microbial compositions could be purely because several people are inside a capsule over three days. An ideal control would be if the same subjects were housed in the same capsule for three days on earth. Does the "pre-launch" data refer to this control group? I probably missed it if that is the case. Otherwise, I find the data of this project difficult to interpret.

The pre-flight samples refer to the same crew members sampled while not cohabitating in the months leading up to the spaceflight. Detected metagenomic shifts could arise, for example, from the abundance of organisms shifting to above the limit of detection by sequencing or transfer between individuals. Metatranscriptomic shifts do not require microbial transfer – a native species experiencing an increase in expression is all it would take for a detectable signal.

We agree that a parallel ground study would be useful, but that data does not exist at present, and SpaceX could not release an additional unused Dragon capsule to sequence another cohort, but this has been requested for future missions with their flight operations team. However, we do not feel this is a critical control for this study, given the large number of negative and positive controls, what we observe in the discordant shifts between metagenomics and metatranscriptomics across the exact same body sites over time, and because of what is known about spaceflight and the microbiome.

For example, spending time in a tight space does not normally cause herpesvirus activation in control cohorts. But spaceflight does activate herpes strains in 61% of people (<https://www.ncbi.nlm.nih.gov/pmc/articles/PMC6374706/>) in spaceflight. We address this in the Discussion/Supplementary Discussion:

nature portfolio

“Chief among our findings was that native microbiome shifts were highly correlated to host immune changes. Naturally, a microbial shift can affect the host immune system – or vice versa – without the initial cause being “space-specific” (i.e., due to microgravity or radiation). Strain sharing, for example, could be – and likely is – a function of humans sharing close quarters. That said, astronauts have been documented as experiencing immune and viral activation¹⁵; typically, this effect is not attributed solely to cohabitation, and we showed here that strains likely “acquired” from crew members in flight were not associated with immune cell changes. We claim, therefore, it is unlikely strain sharing due to close quarters – or even variable sanitation in-flight – explains the entirety of the link between host immune response and the microbiome.

Future manuscripts, of course, could leverage this dataset as well as data from analog astronaut studies on Earth to more rigorously test this hypothesis.”

In short, a portion of changes may be from cohabitation – but it is unlikely that all of them are, and this dataset is the largest to-date that enables such comparisons, which can help tease out these interactions when we begin to compare to other missions. Further of note, the changes that are almost surely from cohabitation – skin metagenomic shifts of specific taxa – are not correlated to immune cell gene expression changes. On the other hand, metatranscriptomic shifts in the oral microbiome, where there is no known exchange between individuals, are associated with immune cell gene expression changes. We thus claim that these data, combined with the known effects of spaceflight that are uniquely distinct from cohabitation, bolsters our claim that at least some of the changes are likely space-associated. It will, however, require future work to determine which of these shifts are consistently observed or mission-specific.

3. In response to my comment regarding correlation analyses, the authors note on page 46 that they combined data from all sites to construct correlations. Does it mean they treated all sites as one organ for computing correlations? Since these are relative abundance data, the standard concept of Pearson correlation is not appropriate. Did they use SPARCC or some of the other recent methods? From the heatmaps, I am puzzled that all correlations among the bacterial families are all positive. Why is that? Is it an artifact of the correlation tool used? Or is there a real biology?

We agree this analysis is worthy of a deeper dive than what we did; however, we are attempting in these revisions to balance requests for a shorter, tighter manuscript with additional analytic goals. Previously, this figure was referenced in one line in the results, and we feel the analysis requested would require (and warrant) much more text. Therefore, we have opted instead to remove this figure and recommend correlational network-based analyses as follow-ups in following work. We feel it is appropriate, as it did not change the main results of the paper (as highlighted by Reviewer 2) and is not the focus of the main results, but we have cited SPARCC in the discussion to highlight future directions and analytical approaches.

4. It is true that linear models have a history in microbiome data analysis, but recently developed algorithms such as LinDA would be better alternatives to deal with compositionality in the microbiome data. There is a risk of high false discovery rates when compositionality is ignored.

We agree that linear models have some limitations, and we have added to our revised paper a battery of methods to help reduce risk of analysis error. As the reviewer keenly notes, there are a number of alternative statistical/transformation approaches that we could try, but we believe our current comparison of methods is appropriate, given the standard in the microbiome field. Moreover, since we've used 11 methods in this paper and seen similar results across all of them, we believe that using LinDA to adjust for compositionality bias is a good idea, but one that should be robustly compared across a wide range of sample types in a future study (we now

cite LinDA in the manuscript as a potential additional analytic approach). We claim our concordance across transformation methods accounts for a degree of bias in compositional data, though in future work on this dataset, we will perform even more comparisons with additional methods. We address this in our Discussion section to highlight this for readers.

We are grateful and very appreciative of the reviewers for their careful, thorough, and helpful reviews and hope they find the new manuscript satisfactory.

Appendix:

Supplementary Discussion

Specific human genes and their association with microbiome features

An additional manuscript focuses more on the host-side of immune activation (Kim and Tierney et al, in review, *Nature*), reporting specific human genes that seem to be associated with microbial features and integrating additional datasets. However, for completeness, we briefly document here some human genes of interest that were microbiome-associated. By cell type, we documented the most strongly associated genes with microbial features (Supplementary Table 10). For bacteria, gene functions were annotated with, for example, long non-coding RNAs (across all cell types), immunoglobulin genes (CD14 monocytes), and interferon regulatory factors. We additionally uncovered associations with specific immune modulatory genes such as CXCL10, XCL1, CXCL8 (immune cell migration), NLRC5, HLA genes, CD1C (antigen presentation/co-stimulation), SLC2A9 (immune cell metabolism), IRF1, NR4A3, STAT1 (transcription factors that specify immune cell states) that increased across multiple immune cell types (B cells, CD4 T-cells, CD8 T- cells, CD14 monocytes, DCs, Natural Killer (NK) cells).

Further notes on the complexity of viral classification

In this study, we attempted to measure viral shifts as a function of flight as well. Measuring viral abundances in metagenomic and (particularly) metatranscriptomic data is extremely challenging. First, the decontamination process we used to remove environmental contaminants was not designed for organisms as ubiquitous and with such short, diverse genomes as viruses. Likely, we are removing many true positive viruses from our dataset. Second, we are also working with low biomass samples; it is hard enough to detect bacteria with high confidence, but viruses are another challenging matter entirely.

Finally, and most importantly, taxonomic classification via short read alignment to viral genomes is deeply imperfect. We observed, for example, many significant viruses shifting with unusual names, often mapping to viral species not typically found in or on humans. Simply put, the viral universe is so vast that we expect these alignments are likely due to a combination of spurious read mapping and databases lacking the strains that are truly present. As a result, we are only able to work with state-of-the art methods, comparing the results across taxonomic ranks.

We are confident in the results highlighted in the main text of this manuscript regarding viruses: mostly transient, pan-body shifts in-flight for both prokaryotic and eukaryotic viruses, some of which are correlated to immune cell gene expression changes. We compared multiple methods, finding concordant results, and we benchmarked the method used in the main text (Xtree alignment to GenBank) via rigorous testing on synthetic genomes (Supplementary Figure 18), finding accurate alignment at the genus level. Further, we identified viral genes as shifting in-flight in the functional/gene catalog analysis (Supplementary Figures 14-17). That said, we are also certain that some of the taxonomic classifications are not accurate, so any atypical clades highlighted in the regression results (Supplementary Table 9) should be considered with appropriate caution.

Other drawbacks of this study and future opportunities

There are several opportunities to expand upon this work in future studies and missions. Analytically, our lasso-based approach for immune-microbe-interaction modeling immune cell gene expression changes does not inherently allow for statistical inference or account for inter-individual variation. Further, some of our samples had very low biomass, requiring PCR-amplification (18 cycles) for RNA-sequencing data, which can increase duplicate rates of sequences. For this reason, we attempted to take a conservative and systematic modeling approach to our effort. Specifically, 1) we implemented multiple algorithms and compared their concordance, 2) set coverage thresholds for bacterial and viral taxa to filter probable false positives, 3) used multiple, state-of-the-art taxonomic classifiers and compared our findings among all of them, and 4) implemented and compared both generalized linear models and mixed effect models, bearing in mind that the latter can face interpretability challenges with smaller sample sizes. Additional modeling strategies, including network analyses², could be implemented in addition to those we have tested here. For example, recently developed methods for controlling false positive rates in compositional data could be potentially useful for this manner of microbiome data¹.

We additionally used 76 negative controls to attempt to avert false positive signals, which can stem from contamination and the kitome. However, this approach is far from perfect and likely removes present organisms. Depending on their aim, future studies should alter collection methods to increase the amount of biomass collected sampling (e.g., using one swab for multiple skin sites) or examine relatively unbiased methods of amplification³. They additionally should encourage more detailed reporting on diet and cleaning methods (e.g., wet wipes) to adjust for potential confounders introducing foreign microbial DNA into the host.

Additional experiments and missions can further test a microbiome-derived theory of spaceflight-associated immune c changes. In addition to stress-testing our findings and increasing sample sizes, future spaceflight studies should consider several enhancements. For instance, they should compare sequestered ground controls to discern differences between space-driven and proximity-driven immune shifts. Additionally, future efforts should design experiments that enable a deeper view into the causality of microbe immune associations rather than just noting their existence. Exploring some of these hypotheses through animal or organoid models could be valuable.

1. Zhou, H., He, K., Chen, J. & Zhang, X. LinDA: linear models for differential abundance analysis of microbiome compositional data. *Genome Biol.* **23**, 95 (2022).
2. Friedman, J. & Alm, E. J. Inferring correlation networks from genomic survey data. *PLoS Comput. Biol.* **8**, e1002687 (2012).

3. Ahsanuddin, S. et al. Assessment of REPLI-g Multiple Displacement Whole Genome Amplification (WGA) Techniques for Metagenomic Applications. *J. Biomol. Tech.* **28**, 46–55 (2017).

Decision Letter, first revision:

Message: Our ref: NMICROBIOL-23082268A

1st December 2023

Dear Dr. Mason,

Thank you for submitting your revised manuscript "The microbiome architecture of short-term spaceflight and its potential link to host immune activation" (NMICROBIOL-23082268A). It has now been seen by the original referees and their comments are below. The reviewers find that the paper has improved in revision, and therefore we'll be happy in principle to publish it in Nature Microbiology, pending minor revisions to satisfy the referees' final requests and to comply with our editorial and formatting guidelines.

We will ask for some additional text edits in line with the remaining concerns including toning down statements, discussing limitations and removing or reducing the section on strain sharing. We will send further instructions with the checklist mentioned below.

Thank you again for your interest in Nature Microbiology Please do not hesitate to contact me if you have any questions.

Sincerely,

Reviewer #1 (Remarks to the Author):

The authors have addressed my comments satisfactorily.

2Reviewer #2 (Remarks to the Author):

i have concerns that there are contaminants or read mapping issues with the methods, but i don't have any further suggestions given the samples obtained and processed for this study.

I am still not sure that the authors can make the claim that they are seeing strain sharing based on strainphlan analysis without benchmarking for these species.

no further comments.

Reviewer #3 (Remarks to the Author):

I thank the authors for their careful responses to my previous comments. In response to my concerns regarding the lack of proper negative control on earth, they cited an interesting article (Rooney et al., Front. Microbiol.) describing the effects of spaceflight and the herpesvirus activation. I am still having difficulty inferring that spaceflight is the cause for microbial shifts in the compositions. I am not sure if compared to pre-flight, during the three-day flight, the crew brushed their teeth, showered, changed their sleep patterns and diet. Thus, can the differences be partly due to changes in their daily activities along with staying in close proximity to one another for 3 days, and not entirely due to space flight. Granted space-flight does induce stress, as reported in the reference cited by the authors, but it is only part of the effect. I wonder if similar changes to microbiome will be seen for people taking a non-stop long distance flight from NYC to Australia. Note that on lines 227 – 232, the authors observed a reduction in beta-diversity among the crew members. Perhaps this is due to similar diet, environment, and hygiene practices during the three day space-flight. Comparisons with pre-flight specimens is not very convincing to me that changes are due to spaceflight and not due to behavior and environment.

Final Decision Letter:

Message 9th February 2024

:

Dear Chris,

I am pleased to accept your Article "Longitudinal multi-omics analysis of host microbiome architecture and immune responses during short-term spaceflight" for publication in Nature Microbiology. Thank you for having chosen to submit your work to us and many congratulations.

3Over the next few weeks, your paper will be copyedited to ensure that it conforms to Nature Microbiology style. We look particularly carefully at the titles of all papers to ensure that they are relatively brief and understandable.

Please note that *Nature Microbiology* is a Transformative Journal (TJ). Authors may publish their research with us through the traditional subscription access route or make their paper immediately open access through payment of an article-processing charge (APC). Authors will not be required to make a final decision about access to their article until it has been accepted. Find out more about Transformative Journals

Authors may need to take specific actions to achieve compliance with funder and institutional open access mandates. If your research is supported by a funder that requires immediate open access (e.g. according to Plan S principles) then you should select the gold OA route, and we will direct you to the compliant route where possible. For

authors selecting the subscription publication route, the journal's standard licensing terms will need to be accepted, including self-archiving policies. Those licensing terms will supersede any other terms that the author or any third party may assert apply to any version of the manuscript.

With kind regards,